# Several explorations on how to construct an early warning system for local government debt risk in China

**Xing Li**[1], **Xiangyu Ge**[2]*, **Cong Chen**[3]

**1** Experimental Teaching Centre, Hubei University of Economics, Wuhan, Hubei, China, **2** School of Statistics and Mathematics, Zhongnan University of Economics and Law, Wuhan, Hubei, China, **3** School of Finance and Economics, Jiangsu University, Zhenjiang, Jiangsu, China

* xiangyu_ge@163.com

**Data Availability Statement:** Li, Xing, 2022, "Data and Program for ""Several Explorations on How to Construct an Early Warning System for Local Government Debt Risk in China"", https://doi.org/10.7910/DVN/OMAQMM, Harvard Dataverse, V1.

## Abstract

This paper aims to explore several ways to construct a scientific and comprehensive early warning system (EWS) for local government debt risk in China. In order to achieve this goal, this paper studies the local government debt risk from multiple perspectives, i.e., individual risk, contagion risk, static risk and dynamic risk. Firstly, taking China's 30 provinces over the period of 2010~ 2018 as a sample, this paper establishes early warning indicators for individual risk of local government debt, and uses the network model to establish early warning indicators for contagion risk of local government debt. Then, this paper applies the criteria importance though intercrieria correlation (CRITIC) method and coefficient of variation method to obtain the proxy variable I, which combines the above two risks. Secondly, based on the proxy variable I, both the Markov-switching autoregressive (MS-AR) model and coefficient of variation method are used to obtain the proxy variable II, which comprehensively considers the individual risk, contagion risk, static risk and dynamic risk of local government debt. Finally, machine learning algorithms are adopted to generalize the EWS designed in this paper. The results show that: (1) From different perspectives of local government debt risk, the list of provinces that require early warning is different; (2) The support vector machines can well generalize our EWS.

## 1. Introduction

Nowadays, the growing public debt and on-going debt crisis in many countries, such as the US, China, Spain, Italy, Brazil and Mexico, are alarming policymakers and scholars. Similarly, ever-growing sovereign debt at sub-national levels like US states and China local governments as well as at the municipality level has caused worldwide concern. Among the highly indebted countries, China is gaining particular attention. This is because since China granted local governments the right to issue bonds in 2009, the balance of local government debt in China has surprisingly increased by 200% in a short decade, i.e., from 10717.491 billion yuan in 2010 to 21330 billion yuan in 2019 (China's National Bureau of Statistics and National Audit Office). Such a rapid growth has spurred an academic debate on how to strengthen the supervision of public debt and how to avoid the outbreak of debt crisis. As a result, the construction of early

**Funding:** The authors disclosed receipt of the following financial support for the research design, data collection and analysis, authorship, and/or publication of this manuscript: Xiangyu Ge acknowledges the financial support provided by the National Natural Science Foundation of China (Grant No. 71974204). Xing Li acknowledges the financial support from the Doctoral Scientific Research Startup Fund for the Hubei University of Economics (Grant No. XJ21BS12).

**Competing interests:** The authors declare that there are no competing interests regarding the publication of this paper.

warning system (EWS) for public debt risk has become one of the research hotspots in academia.

An effective EWS can help governments to assess the local government debt risk and take timely measures to prevent the risk. Therefore, scholars have adopted various models to construct the EWS, such as the stochastic model [1–3], general equilibrium model [4] and econometric model [5, 6]. Although the existing EWSs have great enlightenments in assessing and predicting local government debt risk, further questions need to be explored. Do the existing EWSs consider various types of local government debt as much as possible? Do they take the contagion risk of local government debt into account? Do they explore the regime-switching risk of local government debt? In view of these questions, this paper takes China's 30 provinces over the period of 2010 to 2018 as a sample, and intends to go deep into the EWS for local government debt risk.

The marginal contributions of this paper are as follows: (1) Besides the explicit local government debt, the implicit local government debt will be included in this paper, making our EWS cover more types of local government debt. (2) Based on the spatial spillover effect of China's local government debt risk [7–9], the contagion risk of local government debt will not be ignored in our EWS. (3) Rather than using the subjective weighting methods, the objective weighting methods (the criteria importance though intercrieria correlation method and coefficient of variation method) will be used to determine the weight of early warning indicators, thus ensuring the fairness of our indicator weighting. (4) Considering that the data of local government debt risk of China's each province are typically non-linear time series and the Markov-switching autoregressive (MS-AR) model can effectively capture different regimes of time series data [10–12], the MS-AR model will be used to study the static risk and dynamic risk of local government debt for each province. (5) The machine learning algorithms will be introduced to debug an optimal classifier to generalize the EWS, thus improving the convenience and practicality of our EWS. However, because of the small data size, it is necessary to follow the changes in China's local government debt risk and update the sample data in future research.

The rest of the paper is organized as follows. Section 2 provides the literature review. Section 3 describes the construction idea of our EWS. Based on the spatial spillover effect of China's local government debt risk, Section 4 calculates the proxy variable I which combines the individual risk and contagion risk of local government debt, and obtains the proxy variable II which comprehensively considers the individual risk, contagion risk, static risk and dynamic risk of local government debt. Section 5 debugs an optimal classifier based on machine learning algorithms and Section 6 concludes.

## 2. Literature review

As western countries granted governments the right to raise debt earlier than China, western scholars made pioneering attempts in constructing the EWS. The study on the EWS for public debt can be tracked back to 1980s [13], and has sprung up in the US, Italy and Colombia. Recently, new models and new techniques are constantly introduced into enhancing the function of EWS, and the research object is extending from sovereign debt risk to sub-national debt risk.

Among the existing EWSs constructed by western scholars, the multinomial logistic regression belonging to the econometric models has been the most widely used. Pioneering studies by Ciarlone and Trebeschi [14] and Fuertes and Kalotychou [15] have applied the multinomial logistic regression to estimate the sovereign debt risk of one country and several countries, respectively. Since then, following in the footsteps of Ciarlone and Trebeschi [14] and Fuertes and Kalotychou [15], western scholars have repeatedly confirmed the predictive value of multinomial logistic regression, such as Fournier and Bétin [16] (see Table 1). Besides the multinomial logistic regression, other models have enriched the construction of EWS. For example,

the financial particle theory and objective decomposition method have been comprehensively used by Turay et al. [17] to develop an EWS for local government debt risk. The parametric proportional hazards regression, conventional logistic regression and Bayesian model averaging have been employed by Kamra [18] to explain the uncertainty of EWS for debt crisis in emerging markets. A new regression-tree based approach along with a signal model has been adopted by Savona and Vezzoli [19] to achieve a balance between in- and out-of-sample predictability of sovereign defaults. A more powerful dynamic-recursive early warning model has been established by Dawood et al. [20], who sends more accurate out-of-sample warning signals of sovereign debt crisis.

With the development of software and econometric models, a new trend in the latest work is to introduce the machine learning algorithms into the EWS for municipal debt risk. Representatives are Antulov-Fantulin et al. [21], Alaminos et al. [22] and Zahariev et al. [23]. To be specific, an optimal machine learning model has been selected by Antulov-Fantulin et al. [21] from the gradient boosting machine, random forest, lasso and neural network, finding that the gradient boosting machine performs the best in predicting bankruptcy of local government in Italy. The fuzzy decision trees, AdaBoost, extreme gradient boosting and deep learning neural decision trees all have been proved a good early warning performance for sovereign debt crisis by Alaminos et al. [22]. The support vector machine (SVM) has been confirmed applicable to model the dependence of the debt ratio of Italy and Greece by Zahariev et al. [23].

Faced with less than 15 years of local government debt data, Chinese scholars still made valuable explorations on the construction of EWS. Representatives are the BP neural network by Shi et al. [24] and Hong and Liu [25], the principal component analysis along with a multivariate discriminate analysis by Tao [26], the CRITIC method combined with the grey relational analysis and TOPSIS method by Liu and Lu [27], the TOPSIS method with the Delphi method and SVM algorithm by Li et al. [28], the fuzzy evaluation by Xu et al. [29], the analytic hierarchy process (AHP) with an entropy method by Shen and Jin [30], and the grey prediction model along with the theory of risk energy release by Gao and Zhang [31].

Although previous studies have great enlightenments in constructing EWS for public debt risk, there are still some research gaps at the local government debt level, especially for China's local government debt risk. The research gaps are as follows: (1) In terms of local government debt statistics, previous studies mainly construct the EWS for explicit debt of local government; and limit the research object to local government bonds [1, 4, 29, 32], or debt from local government financial vehicles [26, 27], or both the two ones [17, 28, 30, 31, 33, 34]. (2) In terms of early warning indicators, previous studies only select indicators from single and static perspectives. Each local government is regarded as a single debt risk carrier. Both the contagion effect and the probability of deterioration or turnaround of debt risk are not taken into account. Although few studies (see Wang and Chen [35]) have considered the regime-switching risk of local government debt, the contagion risk of local government debt is unfortunately ignored. (3) In terms of weighting methods, previous studies tend to use subjective methods or a subjective and objective integrated methods [25, 27–30]. Decision maker needs to compare the criteria subjectively, thus reducing the objectivity of conclusions. (4) In terms of machine learning algorithms, there exists the following problems for China's case: 1) The applicability of machine learning algorithms for China's small sample data. Fundamentally, many machine learning algorithms have requirements for sample size. Taking the BP neural network model as an example, it is likely to fall into the trap of local optimum when applied to small sample. 2) How to optimize the parameters of machine learning algorithms and avoid overfitting in face of China's small sample data. Although the SVM has proved a good performance in generalizing the EWS for government debt risk by western scholars, e.g., Alaminos et al. [22] and Zahariev et al. [23], the SVM has been deliberately simplified when applied to China's case. Some scholars

**Table 1. The previous EWSs and our EWS.**

| Representatives | Model/ Technique | Contribution | Limitation |
|---|---|---|---|
| **Ciarlone and Trebeschi [14]** | Multinomial logistic regression | It discovers which macroeconomic factors determine the debt crisis, and obtains crisis signals from the default parameters. | Due to the limited sample size, the two aspects of EWS for debt crisis are not well integrated. |
| **Fuertes and Kalotychou [15];** | | It controls for country, regional and time heterogeneity to improve the forecast power of sovereign default models. | The EWS is based on binary dependent variable model, with an inherent endogeneity problem. |
| **Fournier and Bétin [16]** | | It complements the literature on debt limits in advanced economies. | The sample does not include low-income countries. |
| **Turay et al. [17]** | Financial particle theory+ Objective decomposition method | It constructs a comprehensive EWS including "borrowing", "usage" and "repaying" early warning indicators. | The Delphi method (a subjective weighting method) affects the objectivity of weights of indicators. |
| **Kamra [18]** | Parametric proportional hazards regression+ Conventional logistic regression+ Bayesian model averaging | It expands the data coverage to emerging markets, and uses alternate empirical techniques to select early warning indicators. | The EWS is too complex. The warning speed and convenience need to be improved. |
| **Savona and Vezzoli [19]** | Regression-tree based approach + Signal model | It reports different risk indicators used for four main purposes, and helps local governments to deal with exogenous financial crises. | Any claim of generalizability beyond the reviewed material needs to be further verified. |
| **Dawood et al. [20]** | Dynamic-recursive early warning model | It proposes a different specification of crisis variable, and develops a more powerful dynamic-recursive forecasting technique. | The sample data of developing countries are very limited. |
| **Antulov-Fantulin et al. [21]** | Machine learning algorithms | It enriches the literature on EWS by using the recent machine learning algorithms. | More explanations about the role which social, human and cultural capital play in the municipal defaulting process need to be elaborated. |
| **Alaminos et al. [22]** | | It shows the superiority of computational techniques over statistics in terms of the precision in the EWS for sovereign debt risk. | The country strength model and financial strength model need to be modified to increase the generalization of EWS. |
| **Zahariev et al. [23]** | | It constructs a good visible EWS by using Python software. | The selection of early warning indicators needs to be elaborated. |
| **Shi et al. [24]** | BP neural network | It fills the research gap in the EWS for China's government financial risk. | The BP neural network has slow convergence speed and low training efficiency, and tends to fall into the trap of local optimum due to the small sample size. |
| **Hong and Liu [25]** | | It adopts a subjective and objective integrated method in the indicator weighting, and constructs a nonlinear simulation EWS for China's local government debt risk. | The subjectivity of AHP [a] method. |
| **Tao [26]** | Principal component analysis + Multivariate discriminate analysis | It assesses the default risk and debt-bearing boundaries of local governments in China. | The sample is limited to implicit debt of local government. |
| **Liu and Lu [27]** | CRITIC method + Grey relational analysis+ TOPSIS method | It uses the CRITIC [b] method to weight indicators. | The weight derived from TOPSIS [c] method leads to the problem of pseudo weight. The sample is limited to implicit debt of local government. |
| **Li et al. [28]** | TOPSIS method+ Delphi method + SVM algorithm | It introduces machine learning algorithms into the construction of EWS for China's local government debt risk. | The subjectivity of TOPSIS and Delphi method. The penalty parameter $C$ is ignored in the parameter optimization of SVM. |
| **Xu et al. [29]** | Fuzzy evaluation | It uses the AHP and fuzzy comprehensive evaluation to construct the EWS for China's H province. | The subjectivity of AHP and fuzzy evaluation method. The implicit debt is ignored in the sample. |
| **Shen and Jin [30]** | AHP method+ Entropy method | It selects indicators based on the inherent logic of local government debt risks, and constructs a multi-level EWS. | The subjectivity of AHP method. |
| **Gao and Zhang [31]** | Grey prediction model+ Theory of risk energy release | It constructs a dynamic model of local government debt risk assessment by Vensim PLE software, and obtains a grey prediction model. | The program design is too complicated, with slow operation speed. |

*(Continued)*

**Table 1.** (Continued)

| Representatives | Model/ Technique | Contribution | Limitation |
|---|---|---|---|
| **Our EWS** | CRITIC method+ Coefficient of variation method+ MS-AR model + Machine learning classifier | It constructs a comprehensive EWS for China's local government debt risk, and generalizes the EWS by machine learning algorithms. | The sample size is limited. |

[a] AHP: analytic hierarchy process.

[b] CRITIC: criteria importance though intercrieria correlation.

[c] TOPSIS: technique for order preference by similarity to an ideal solution.

only optimize the parameter *gamma* in Gaussian kernel function of SVM, without considering the penalty parameter *C* (see Li et al. [28]). 3) Application reason is not convincing. Different from the real cases of local government bankruptcy in western countries, China has not witnessed a case of local government bankruptcy due to some policy factors. Therefore, Chinese scholars cannot define the local governments as default or non-default by machine learning classifiers like western scholars. From the few attempts made by Hong and Liu [25] and Li et al. [28], the machine learning algorithms is adopted to extract experts' subjective experience- because the subjective weighting methods such as AHP and Delphi are widely used in their studies. However, whether the experts' experience is valuable and professional enough has not been demonstrated, making their conclusions less interpretable and objective.

In view of the above research gaps, this paper intends to construct an EWS from multiple perspectives of local government debt risk, and debugs a machine learning classifier to generalize the EWS. We believe that our EWS is of some reference significance to China and other countries with similar risk problems. That is, China and other countries can learn from the ideas of this paper to construct a comprehensive EWS to quickly grasp the local government debt risk signals.

## 3. Construction ideas

### 3.1. Conceptual framework

There are four types of local government debt risks in this paper:

*The individual risk of local government debt*: due to the fact that local government as a subnational administrative agency cannot easily go bankruptcy, the individual risk of local government debt can be defined as "too big to fail" of financial risk. Although the Ministry of Finance and the People's Bank of China are exploring the local government bankruptcy, financial institutions and the public still adhere to the firm beliefs in the government. Then, the moral hazard of "too big to fail" encourages an aggressive expansion of local government debt, thus accumulating a great amount of local government debt risk. Nowadays, scholars have attached great importance to the individual risk of local government debt, and established various EWSs to predict the risk. Representatives are Wijayanti and Rachmanira [6], Turay et al. [17], Hong and Liu [25], Li et al. [28], Gao and Zhang [31], Jin et al. [36] and Zhang [37].

*The contagion risk of local government debt*: it can be defined as "too tightly correlated to fail" of financial risk. Referring to the empirical conclusions of Li et al. [9], there exists spatial correlations and spillover effect of local government debt risks in China's 30 provinces. Such spatial characteristics are likely to cause contagion of local government debt risks between provinces.

*The static risk of local government debt*: it refers to how risky the local government debt is when it maintains its original risk state. As the local government debt risk is likely to transit from low-risk state to high-risk state in some China's provinces (see Wang and Chen [35]), it is necessary to pay attention to the state changes of local government debt risk. Obviously, if the local government debt of a province is in high-risk state for a long time, the local government debt

risk of that province is greater; conversely, if the local government debt of a province is in low-risk state for a long time, the local government debt risk of that province is smaller.

*The dynamic risk of local government debt*: it describes the regime-switching risk of local government debt when it switches between low-risk state and high-risk state. If the local government debt of a province switches from a low-risk state to a high-risk state, the province is experiencing a deterioration of local government debt risk; conversely, if the local government debt of a province switches from a high-risk state to a low-risk state, the province is experiencing an easing of local government debt risk.

### 3.2. Hypothesis

Risks are not single and static, but contagious and dynamic. The contagion effect and dynamics can be described by empirical models. Regarding the contagion risk, Bianchi et al. [38] take the network structure perspective and use the standard eigenvector centrality to model contagion in financial market; Anagnostou et al. [39] incorporate contagion in portfolio credit risk model by using network theory; Berloco et al. [40] use the network model to capture firms' fragility to shocks. Regarding the dynamic risk, Jutasompakorn et al. [41] identify the banking crisis dates via the MS-AR model; Xaba et al. [42] explore the performance of MS-AR model to forecast the quarterly exchange rate of South Africa; Makatjane and Kagiso [43] realize a dynamic early warning of the Johannesburg stock exchange all-share index through a two regime MS-AR model. Referring to the above practices, this paper will apply the network model and MS-AR model to construct a comprehensive EWS local government debt risk in China. Based on the above, this paper proposes the following hypothesis:

Hypothesis 1: A proxy variable that comprehensively considers the individual risk, contagion risk, static risk and dynamic risk of local government debt of each province can be obtained by network model and MS-AR model.

In risk research, there is a heated discussion on foundational issues about concepts and perspectives. The development of well-founded risk perspectives is a crucial issue to intensify the practice of risk analysis. Fundamentally, different risk perspectives may lead to different early warning results [44]. Taking China's data as sample, Shen et al. [30] construct an EWS from the individual risk perspective, finding that Qinghai, Hunan, Guizhou, Heilongjiang and Jilin have greater local government debt risk. Wang and Chen [35] study from the static and dynamic perspectives, and conclude that Jiangsu, Hebei, Shandong, Chongqing, Xinjiang and Henan provinces have greater local government debt risk. Different from previous studies, this paper has four risk perspectives, i.e., individual risk, contagion risk, static risk and dynamic risk. Such multiple risk perspectives may lead to different early warning results. Thus, this paper proposes the following hypothesis:

Hypothesis 2: Different risk perspectives may lead to different list of provinces that require early warnings.

Moreover, due to the numerous advances in software, recent works increasingly use the machine learning algorithms to finalize prediction. Scholars have showed how machine learning algorithms outperform the traditional econometric models, and recognized the superiority of machine learning algorithms in constructing EWS (see Antulov-Fantulin et al. [21], Alaminos et al. [22] and Zahariev et al. [23]). Based on this, this paper will use machine learning algorithms to improve the generalization performance of our EWS. Therefore, the following hypothesis is proposed:

Hypothesis 3: An optimal machine learning classifier can be debugged to generalize our EWS.

## 3.3. Research design

The research design is shown in Fig 1. Firstly, early warning indicators of individual risk and contagion risk of local government debt will be selected, respectively. Secondly, both the CRITIC method and coefficient of variation method will be used to calculate the proxy variable I of each province. The proxy variable I reflects the individual risk and contagion risk of local government debt. Thirdly, based on the proxy variable I, the MS-AR model will be applied to investigate the regime-switching risk of local government debt for each province. Then, the proxy variable II which comprehensively considers the individual risk, contagion risk, static risk and dynamic risk of local government debt can be obtained by the MS-AR estimation. Finally, machine learning algorithms will be introduced to debug an optimal classifier to generalize our EWS.

## 4. Methodology

Due to the availability of data, this paper chooses a provincial-level unit as the research area [45–48]. Additionally, in order to cover as many types of local government debt as possible, this paper classifies the local government debt into explicit debt and implicit debt according to Polackova's definitions [49]. Then, referring to the statistical caliber of Mao and Huang [50] and Wang [51], the explicit debt is composed of local government bonds and debt re-loans, and the implicit debt is composed of urban investment bonds.

## 4.1. The proxy variable I

**4.1.1. Early warning indicators of individual risk.** Early warning indicators of individual risk of local government debt are detailed in Table 2. The data come from the Wind

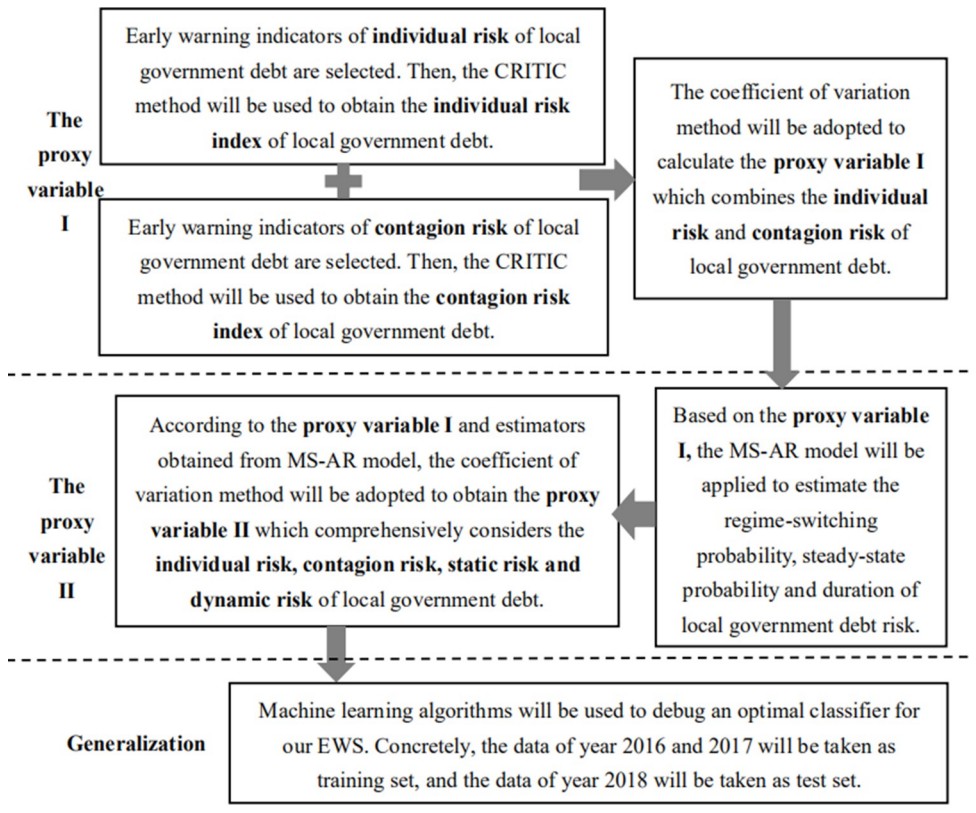

**Fig 1. Research design.**

Table 2. Early warning indicators of individual risk of local government debt.

| Risk chain | Early warning indicators | Measurement | Mild warning | Moderate warning | Severe warning | Attributes | References | Weights by the CRITIC method |
|---|---|---|---|---|---|---|---|---|
| Debt Status | Dependence Degree of Debt X1 | $\dfrac{\text{Current new debt}}{(\text{Current fiscal expenditure}+\text{Debt principal repayment and interest in the current year})}$ | (-∞, 20%) | [20%, 45%] | (45%, +∞) | Positive | Pei and Ouyang [52] | 10.87% |
| | Debt Ratio X2 | $\dfrac{\text{Debt principal repayment and interest in the current year}}{\text{Current GDP}}$ | (0, 10%) | [10%, 25%] | (25%, +∞) | Positive | Li et al. [28] | 8.68% |
| | New Debt Ratio [a] X3 | $\dfrac{\text{Current new debt}}{\text{Current fiscal revenue increment}}$ | (0, 15%) | [15%, 20%] | (20%, +∞) | Positive | Zhang [37] | 3.50% |
| Debt Usage | The Ratio of GDP Growth to Debt Growth [b] X4 | $\dfrac{\text{Current GDP growth}}{\text{Current debt balance growth}}$ | (100%, +∞) | [60%, 100%] | (0, 60%) | Negative | Jin et al. [36] | 4.78% |
| | Cost of Debt X5 | $\dfrac{\text{Interest in the current year}}{\text{Debt balance in the previous year}+0.5\times\text{Current new debt}}$ | (0, 5.93%) | [5.93%, 24%] | (24%, +∞) | Positive | Gao and Zhang [31] | 17.93% |
| Debt Repayment | Debt Servicing Ratio X6 | $\dfrac{\text{Debt principal repayment and interest in the current year}}{\text{Current fiscal revenue}}$ | (0, 15%) | [15%, 30%] | (30%, +∞) | Positive | Gao and Zhang [31] | 10.24% |
| | Debt Burden Ratio X7 | $\dfrac{\text{Current debt balance}}{\text{Current fiscal revenue}}$ | (0, 20%) | [20%, 50%] | (50%, +∞) | Positive | Gao and Zhang [31] | 9.38% |
| | Asset-liability Ratio X8 | $\dfrac{\text{Current debt balance}}{\text{Current asset balance}}$ | (0, 30%) | [30%, 70%] | (70%, +∞) | Positive | Jin et al. [36] | 12.10% |
| Economic Strength | GDP Growth X9 | $\dfrac{\text{Current GDP}-\text{GDP in the previous year}}{\text{GDP in the previous year}}$ | (7%, +∞) | [0, 7%] | (-∞, 0) | Negative | Gao and Zhang [31] | 4.40% |
| | Deficit Ratio X10 | $\dfrac{\text{Current fiscal deficit}}{\text{Current GDP}}$ | (0, 1%) | [1%, 3%] | (3%, +∞) | Positive | Li et al. [28] | 14.10% |
| | Change of Fiscal Revenue and Expenditure [c] X11 | $\dfrac{\text{Current fiscal revenue growth}}{\text{Current fiscal expenditure growth}}$ | (100%, +∞) | [50%, 100%] | (0, 50%) | Negative | Gao and Zhang [31] | 4.02% |

[a] For negative values of X3: if a province's current new debt > 0 with current fiscal revenue increment > 0, the province will be classified in the severe warning area; if a province's current new debt < 0 with current fiscal revenue increment > 0, the province will be classified in the mild warning area.

[b] For negative values of X4: if a province's current GDP growth > 0 with current debt balance growth < 0, the province will be classified in the mild warning area; if a province's current GDP growth < 0 with current debt balance growth > 0, the province will be classified in the severe warning area.

[c] For negative values of X11: if a province's current fiscal revenue growth > 0 with current fiscal expenditure growth > 0 with current fiscal revenue growth > 0, the province will be classified as in the mild warning area; if a province's current fiscal revenue growth < 0 with current fiscal expenditure growth > 0, the province will be classified in the severe warning area.

database and the empirical results of Li et al. [9]. However, Li et al. [9] do not provide the statistics of principal repayment and interest payable for local government bonds, urban investment bonds and debt re-loans. This paper supplements the relevant calculation as follows:

For the local government bonds: (1) The principal repayment of local government bonds of one province at the end of year *t* equals the sum of matured local government bonds of that province in the *t* year. (2) Since each bond has its own ways of paying interest, terms in the *t* year, interest rates and amounts; the interest of local government bonds should be calculated one bond by one bond for each province. Then, these interests will be aggregated as the total interest of that province at the end of year *t*. The calculation are conducted by Stata 14.0.

For the urban investment bonds: the debt principal repayment and interest are calculated in the same way as those of local government bonds.

For the debt re-loans: referring to the Notice (No. 479[1999]) of "the Ministry of Finance on Certain Issues Concerning the Repayment of Principals and Interests of Debt Re-loans": (1) The principal repayment of debt re-loans of one coastal developed province at the end of year *t* is measured by the sum of debt re-loans obtained in the *t*−6 year. The total interest of that province in the *t* year equals the sum of debt re-loan balance and principal repayment multiplied by an annual interest rate of 5.5%. (2) The principal repayment of debt re-loans of one central or western province at the end of year *t* is measured by the sum of debt re-loans obtained in the *t*−10 year. The interest payable of that province in the *t* year equals the sum of debt re-loan balance and principal repayment multiplied by an annual interest rate of 5%.

**4.1.2. Early warning indicators of contagion risk.** According to social network theory, centrality can be used to express the importance of edges and nodes. The degree of influence of the node on other nodes can be estimated by centrality analysis [53].

Based on this, the centrality indicators from Li et al. [9] will be used to measure the contagion risk of local government debt, as shown in Table 3. However, the mild warning, moderate warning and severe warning are not distinguished in Table 3. In order to determine the warning degree of each province, these indicators will be divided into three equal shares in terms of the year and value. The division rules are: (1) For positive indicators, those provinces ranking in the top 1/3 will be classified in the severe warning area, those in the bottom 1/3 will be in the mild warning area, and those in the middle 1/3 will be in the moderate warning area; (2) For negative indicators, the opposite is true.

**4.1.3. Individual risk index and contagion risk index.** There exists various weighting methods, i.e., the subjective weighting represented by AHP, the objective weighting represented by CRITIC method and coefficient of variation method, the subjective-objective integrated weighting, and the newly-emerging weighting represented by BWM (Best-Worst Method) [56], SWARA (Simultaneous Evaluation of Criteria and Alternatives) [57], SECA (Simultaneous Evaluation of Criteria and Alternatives) [58] and MEREC (Method Based on the Removal Effects of Criteria) [59]. Considering that subjective weighting incorporates subjective judgments of decision maker while some objective weighting needs fussy work with much calculation, this paper adopts the CRITIC method and coefficient of variation method to give objective weights to indicators.

The CRITIC method extracts information from a decision matrix to determine the objective weights of indicators [60]. The quantification between indicator *j* and other contradicting indicators is represented by $\sum_{i=1}^{n}(1 - r_{ij})$. Where $r_{ij}$ refers to the correlation coefficient between indicator *i* and indicator *j*, $\sigma_j$ serves as the standard deviation of indicator *j*, and *n* is defined as

**Table 3. Early warning indicators of contagion risk of local government debt.**

| Centrality | Measurement | Description | Attributes | Weights by the CRITIC method |
|---|---|---|---|---|
| **Degree Centrality** <br><br> **Y1** | $C_D(i) = \sum_{j=1}^{N} a(i,j)$, where $i$ is a given node. If there is a tie between node $i$ and node $j$, then $a(i,j) = 1$. | Degree centrality equals the sum of ties of a given node. The larger the degree centrality of a province is, the more ties the province shares with other provinces in the spatial network of local government debt risk; vice versa. | Positive | 11.51% |
| **Closeness Centrality** <br><br> **Y2** | $C_C(i) = [\sum_{j=1}^{N} d(i,j)]^{-1}$. The farness of a node is the sum of the lengths of the geodesics to every other node. The reciprocal of farness is exactly the closeness centrality. In addition, the larger the farness of a node is, the more marginal the node will be. | Closeness centrality is a measure for a given node that is not controlled by other nodes. According to Freeman [54], the node with the farthest geodesic distance from the central node controls the least information resources, power, prestige and influence. In this paper, the larger the closeness centrality of a province is, the less important the province will be in the spatial network of local government debt risk. | Negative | 10.00% |
| **Betweenness Centrality** <br><br> **Y3** | $C_B(i) = \sum_{j=1}^{N} \sum_{k=1}^{j-1} \frac{g_{jk}(i)}{g_{jk}}$, where $g_{jk}$ equals the sum of geodesics linking node $j$ and node $k$, and $g_{jk}(i)$ represents the sum of geodesics which pass through node $i$. | Betweenness centrality is a measure of the number of times a node occurs on a geodesic, reflecting the degree to which a node controls the flow of information between other nodes. The larger the betweenness centrality of a province is, the stronger ability the province has to promote the spillover effect of local government debt risk. | Positive | 28.50% |
| **Eigenvector Centrality** <br><br> **Y4** | $EC_i = x_i = \lambda^{-1} \sum_{j \in M(i)} x_j = \lambda^{-1} \sum_{j \in N(g)} a_{i,j} x_j$, where $N(g)$ refers to a given network; $g$ stands for the sum of nodes. $A = a_{i,j}$ is the adjacency matrix corresponding to the network $N(g)$; if node $i$ is adjacent to node $j$, then $a_{i,j} = 1$, otherwise $a_{i,j} = 0$. $M(i)$ represents the set of adjacent nodes of node $i$. $\lambda$ illustrates the maximum eigenvalue of $A = a_{i,j}$ by the Perron-Frobenius theorem. | According to the eigenvector centrality, the importance of a node is not only determined by the sum of its adjacent nodes, but also by the importance of the adjacent nodes. Thus, the larger the eigenvector centrality of a province is, the closer the province is to the province of source of contagion. | Positive | 18.42% |
| **Bonacich's Power** <br><br> **Y5** | Given an adjacency matrix $A$, the centrality of node $i$ (described by $ci$) is given by $ci = \Sigma A_{ij}(\alpha + \beta cj)$, where $\alpha$ and $\beta$ are parameters. The centrality of each node is therefore determined by the centrality of the node to which it is connected. In addition, the value of $\alpha$ is used to normalize the measure; and the value of $\beta$ is defined as an attenuation factor, which gives the amount of dependence of each node's centrality on the centralities of the nodes it is adjacent to. | According to Bonacich [55], the greater the Bonacich power of a node, the stronger the node's ability to possess resources through connections with other nodes. In this paper, the greater Bonacich power of a province has, the stronger the contagion ability the province has in the spatial network of local government debt risk. | Positive | 18.55% |
| **Average Reciprocal Distance** <br><br> **Y6** | $ARD(i) = \frac{N-1}{\sum_{j=1}^{N} d(i,j)}$. The average reciprocal distance is the reciprocal of the average geodesic distance of $d(i,j)$. Where $d(i,j)$ refers to the shortest geodesic distance between node $i$ and all other reachable nodes, and $N$ represents the total amount of reachable nodes of node $i$ in the network. | The average reciprocal distance emphasizes the value of nodes in the network. In this paper, the larger the $ARD(i)$ of a province is, the larger probability the province has to accept or transfer the local government debt risks. | Positive | 13.01% |

the number of indicators. Theoretically, the objective weight of each indicator results from the comparison between the importance and contradiction of indicators. Therefore, $C_j$, as the amount of information contained in the indicator $j$, can be expressed as Eq (1):

$$C_j = \sigma_j \sum_{i=1}^{n} (1 - r_{ij}) \tag{1}$$

In Eq (1), the larger the $C_j$ is, the more information the indicator $j$ contains, and the greater weight will be given to indicator $j$. Then, $W_j$, as the objective weight of indicator $j$, can be

expressed as Eq (2):

$$W_j = \frac{C_j}{\sum_{i=1}^{n} C_i} \tag{2}$$

Before the application of CRITIC method, the early warning indicators in Tables 2 and 3 require data processing in the SPSSAU software. Considering that the indicators in Tables 2 and 3 are both positive and negative, the negative ones should take the "reciprocal" to be consistent with the positive ones. In addition, since the dimensions of indicators are different and not comparable, all of the indicators should be "standardized". After the data processing, the CRITIC method can be used to obtain the weights, as shown in the last column of Tables 2 and 3.

After data processing, the values of 0, 1 and 2 will be respectively assigned to the indicators in the mild warning area, moderate warning area and severe warning area. Then, the values will be multiplied by corresponding CRITIC method weights, to obtain the individual risk index and contagion risk index for each province. Due to the limited space, the two indexes are presented in Appendices A and B in S1 Appendix (see all appendices at https://doi.org/10.7910/DVN/1EOZXG).

**4.1.4. The proxy variable I: Individual and contagion risk.** "Too big to fail" and "too tightly correlated to fail" are equally important in risk prevention. Regulatory authorities should be vigilant against the two risks at the same time. Therefore, it is necessary to calculate the proxy variable I to combine the above individual risk index and contagion risk index.

The coefficient of variation method is suitable for researchers to conduct a comprehensive evaluation of the research object. The coefficient of variation of the *i*-th index ($i = 1,2,\ldots,n$) is denoted as $CV_i$, shown in Eq (3):

$$CV_i = \frac{\sigma_i}{\bar{x}_i} \tag{3}$$

In Eq (3), $\sigma_i$ represents the standard deviation of the *i*-th index, and $\bar{x}_i$ serves as the mean value of the *i*-th index. The weight $W_i$ of the *i*-th index can be obtained by Eq (4):

$$W_i = \frac{CV_i}{\sum_{k=1}^{n} CV_k} \tag{4}$$

Before the application of coefficient of variation method, the individual risk index and contagion risk index need to be "standardized", in order to solve the dimensional problem. In this case, the "interval" method is adopted by SPSSAU.

After data processing, the weights of individual risk index and contagion risk index are calculated to be 41.27% and 58.73%, respectively. Based on the two weights, the proxy variable I can be obtained, as shown in Table 4. It can be seen from Table 4 that provinces with the larger proxy variable I are Hunan, Jiangsu, Hubei, Shandong and Tianjin provinces (These provinces have repeatedly ranked among the top 5 provinces in terms of the proxy variable I during the sample period. Specifically, Hunan appeared 9 times, Jiangsu 7 times, Hubei 6 times, Shandong and Tianjin each 5 times).

Consequently, on the basis of Table 4, Appendices A and B in S1 Appendix, China's 30 provinces can be classified into four types without regard for static risk and dynamic risk of local government debt: (1) The first type are the provinces with high-risk, i.e., the provinces ranking the top 10 in both the individual risk index and contagion risk index. (2) The second type are the provinces of vulnerability, i.e., the provinces ranking the top 10 in the individual risk index while the bottom 10 in the contagion risk index. (3) The third type are the province with high-correlations, i.e., the province ranking the top 10 in the contagion risk index while the bottom 10 in the individual risk index. (4) The fourth type are the provinces with low-risk,

**Table 4. The proxy variable I (individual risk + contagion risk).**

| Provinces | 2010 | 2011 | 2012 | 2013 | 2014 | 2015 | 2016 | 2017 | 2018 |
|---|---|---|---|---|---|---|---|---|---|
| Anhui | 1.2305 | 0.9160 | 1.2692 | 1.2692 | 1.4532 | 1.4411 | 1.4063 | 1.4817 | 1.5962 |
| Beijing | 0.7304 | 0.7525 | 0.6017 | 0.9952 | 0.8361 | 1.0030 | 1.0081 | 0.9416 | 1.1272 |
| Chongqing | 0.9781 | 0.9781 | 1.0334 | 1.0499 | 1.1944 | 1.1704 | 1.3075 | 1.0442 | 0.9012 |
| Fujian | 0.6046 | 0.6046 | 0.6599 | 0.6433 | 0.6820 | 0.8522 | 0.9228 | 0.8779 | 0.7006 |
| Gansu | 0.5582 | 0.4295 | 0.3044 | 0.2621 | 0.6468 | 0.4073 | 0.4314 | 0.4314 | 0.4298 |
| Guangdong | 0.8833 | 0.8977 | 0.8812 | 0.9365 | 0.9875 | 1.0112 | 1.1116 | 1.0833 | 0.9903 |
| Guangxi | 0.3908 | 0.4074 | 0.3908 | 0.4295 | 0.5201 | 0.5998 | 0.6420 | 0.6602 | 0.6436 |
| Guizhou | 0.3908 | 0.3711 | 0.8107 | 0.5059 | 0.8724 | 1.0056 | 1.1144 | 1.4819 | 1.2782 |
| Hainan | 0.1893 | 0.2400 | 0.1551 | 0.2234 | 0.2234 | 0.2803 | 0.3901 | 0.3569 | 0.4657 |
| Hebei | 1.3592 | 1.3979 | 1.3979 | 1.2305 | 1.0703 | 1.3654 | 1.3640 | 1.5470 | 1.5817 |
| Heilongjiang | 0.2400 | 0.2234 | 0.2234 | 0.2416 | 0.2969 | 0.3634 | 0.4505 | 0.3891 | 0.4329 |
| Henan | 1.3592 | 1.3979 | 1.2305 | 1.2305 | 1.4366 | 1.5162 | 1.6069 | 1.6325 | 1.5680 |
| Hubei | 1.2305 | 1.3979 | 1.3979 | 1.4366 | 1.5106 | 1.5403 | 1.5903 | 1.5454 | 1.5845 |
| Hunan | 1.3979 | 1.3782 | 1.4366 | 1.4366 | 1.5106 | 1.6084 | 1.6851 | 1.6991 | 1.6542 |
| Inner Mongolia | 0.6433 | 0.6599 | 0.8273 | 0.8288 | 0.8675 | 0.9789 | 1.0826 | 0.8532 | 0.9256 |
| Jiangsu | 1.3397 | 1.3563 | 1.3397 | 1.4524 | 1.4973 | 1.5769 | 1.6358 | 1.5909 | 1.5151 |
| Jiangxi | 0.6433 | 1.2305 | 0.6433 | 0.6820 | 0.8494 | 0.9623 | 0.8522 | 1.2368 | 1.0365 |
| Jilin | 0.3908 | 0.3908 | 0.3908 | 0.3908 | 0.2969 | 0.5757 | 0.6179 | 0.4505 | 0.4329 |
| Liaoning | 0.9393 | 0.8806 | 0.9193 | 0.9946 | 0.8828 | 1.0394 | 1.2185 | 0.9795 | 0.7995 |
| Ningxia | 0.4793 | 0.5685 | 0.3391 | 0.4406 | 0.2803 | 0.3917 | 0.4324 | 0.5832 | 0.6270 |
| Qinghai | 0.2953 | 0.2424 | 0.3210 | 0.3044 | 0.3543 | 0.4479 | 0.5427 | 0.5096 | 0.4647 |
| Shananxi | 1.1808 | 1.1808 | 1.2140 | 1.4532 | 1.4366 | 1.6084 | 1.4320 | 1.6325 | 1.5711 |
| Shandong | 1.1940 | 1.3592 | 1.3758 | 1.4145 | 1.3045 | 1.5162 | 1.5929 | 1.6084 | 1.5817 |
| Shanghai | 0.7635 | 0.5464 | 0.7817 | 0.5851 | 0.8022 | 0.6199 | 0.9357 | 0.6872 | 0.9174 |
| Shanxi | 0.8562 | 0.6433 | 0.8694 | 0.8876 | 0.7355 | 0.7333 | 0.9955 | 1.3417 | 1.1625 |
| Sichuan | 0.5260 | 0.5062 | 0.4495 | 0.4883 | 0.4295 | 0.7365 | 0.6585 | 0.9777 | 0.9043 |
| Tianjin | 1.3979 | 1.2305 | 1.3397 | 1.4981 | 1.6054 | 1.6084 | 1.7439 | 1.0519 | 1.0386 |
| Xinjiang | 0.2234 | 0.2566 | 0.2234 | 0.2621 | 0.3361 | 0.4505 | 0.4505 | 0.4415 | 0.3737 |
| Yunnan | 0.2234 | 0.2203 | 0.2400 | 0.2621 | 0.2787 | 0.4174 | 0.5246 | 0.5246 | 0.4237 |
| Zhejiang | 1.1336 | 1.1723 | 1.1723 | 1.2629 | 1.3198 | 1.4437 | 1.4512 | 0.8191 | 1.0242 |

i.e., the provinces ranking the bottom 10 in both the individual risk index and contagion risk index. The classification is shown in Fig 2.

It is found from Fig 2 that the number distribution of the four types is relatively balanced. Regulatory authorities can implement different policies based on the Fig 2. (1) For the first type of provinces, regulatory authorities should try to reduce their local government debt risk as much as possible. (2) For the fourth type of provinces, regulatory authorities can temporarily not list them as the risky provinces; however, they should pay more attention to the dynamic trend of debt risk. (3) For the second and third types of provinces, which are the provinces inconsistent with the severity of individual risk and contagion risk, regulatory authorities do not need to balance these two risks at the same time, thus reducing the difficulty of local government debt management. However, regulatory authorities can still take some risk management measures. For the second type of provinces, regulatory authorities should conduct one-to-one debt governance; for the third type of provinces, regulatory authorities should pay more attention to the competition among the provinces, so as to curb the spatial spillover effect of local government debt risk.

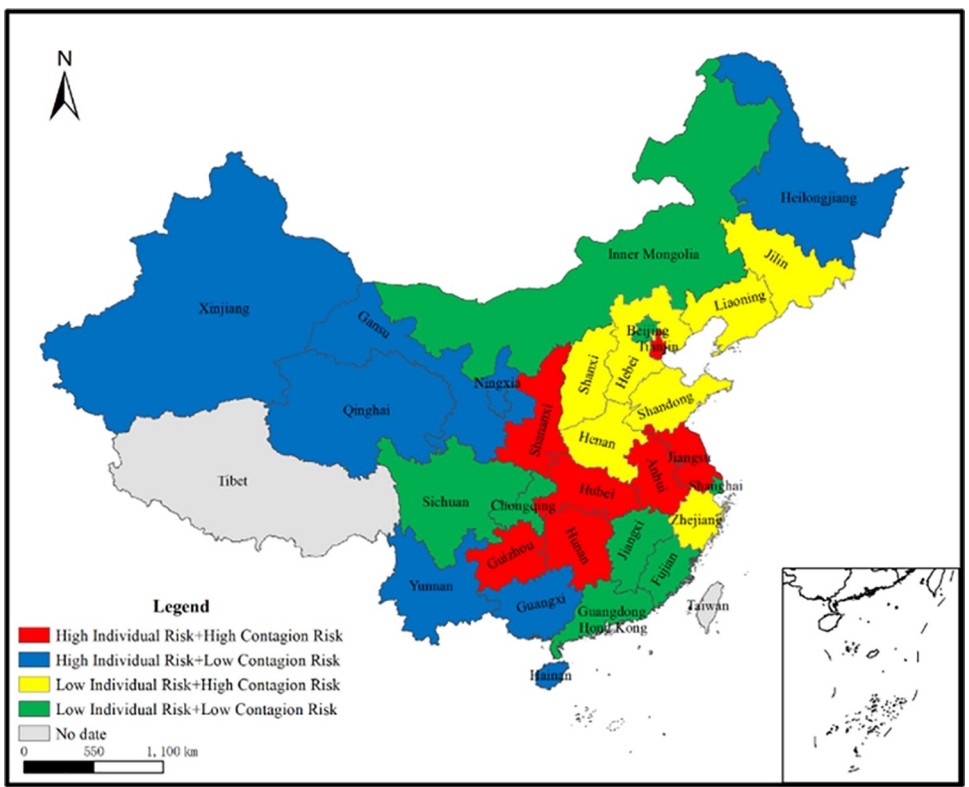

**Fig 2. Four types of local government debt risk: By province; regardless of static risk and dynamic risk of local government debt.**

## 4.2. The proxy variable II

It can be seen from Table 4 that the proxy variable I of each province is not static, but has fluctuations and structural changes during the sample period. Therefore, it is imperative to conduct a dynamic monitoring of China's local government debt risk. Considering that the MS-AR model can effectively capture different regimes of time series data [10–12], this paper applies the MS-AR model to explore the transition probabilities and expected durations of the proxy variable I for each province.

**4.2.1. The MS-AR model.** The MS-AR model is initially introduced by Hamilton [10] to study the regime-switching of market-related time series. The model is shown in Eq (5):

$$
\begin{cases}
x_t = C^{S_t} + \sum_{k=1}^{p} \varphi_k^{S_t} x_{t-k} + \sigma^{S_t}\varepsilon_t,\ S_t \in J, \varepsilon_t : iid.N(0,1) \\
\qquad \Pr(S_t = j | S_{t-1} = 1) = P_{ij} \\
\qquad \sum_{j=1}^{M} P_{ij} = 1, \quad (i,j \in J) \\
\theta = (C^{S_t}, \varphi_k^{S_t}, \sigma^{S_t}, P_{ij} | S_t \in J; k = 1, 2, ..., p; i, j \in J)
\end{cases}
\tag{5}
$$

In Eq (5), $\{x_t\}$ represents a time series, which is characterized by the $p$-order auto-regressive of $M$; $S_t$ refers to a state variable, which is a first-order Markov process with a state space of $J = (1,2,\ldots,M)$; $\theta = (C^{S_t}, \varphi_k^{S_t}, \sigma^{S_t}, P_{ij} | S_t \in J; k = 1, 2, ..., p; i, j \in J)$ denotes a set of undetermined parameters, where the $C^{S_t}$, $\varphi_k^{S_t}$ and $\sigma^{S_t}$ respectively stand for the intercept, coefficient and residual mean square error of the autoregressive model under the state of $S_t$; $\{\varepsilon_t, t = 1,2,\ldots,T\}$ is

defined as a white noise sequence with zero mean following a normal distribution; and $P_{ij}$ describes the one-step transition probability of regime $i$ switching to regime $j$.

Identifying the $P_{ij}$ requires selecting the number of states, i.e., the number of regimes. After many attempts, this paper finds that the two regimes of high-risk state and low-risk state is the best. Therefore, $P_{ij}$ has a total of 2×2 elements in this paper, with the transition probabilities matrix $p$ shown in Eq (6):

$$p = \begin{pmatrix} P(S_t = 1|S_{t-1} = 1) & P(S_t = 2|S_{t-1} = 1) \\ P(S_t = 1|S_{t-1} = 2) & P(S_t = 2|S_{t-1} = 2) \end{pmatrix} = \begin{bmatrix} p_{11} & p_{12} \\ p_{21} & p_{22} \end{bmatrix} \tag{6}$$

According to the conclusion of Hamilton [10], the $\theta$ in Eq (5) can be obtained by using maximum likelihood estimation. Consequently, the conditional logarithmic likelihood function of $\theta$ can be expressed as Eq (7):

$$L(\theta|\Omega_T) = \sum_{t=1}^{T} \ln f(x_t|\Omega_{t-1}; \theta) \tag{7}$$

In Eq (7), $\Omega_{t-1}$ describes the set of all series samples $(x_{t-1}, x_{t-2}, \ldots, x_1)$ observed at $t-1$; $f(x_t|\Omega_{t-1}; \theta)$ equals $\sum_{i=1}^{M} \sum_{j=1}^{M} P_{ij}\xi_{i,t-1}\eta_{j,t}$, where the probability variable $\xi_{i,t-1}$ and the state-mixture density function $\eta_{j,t}$ are defined as Eq (8) and Eq (9), respectively:

$$\xi_{i,t-1} = \Pr(S_{t-1} = i|\Omega_{t-1}; \theta), \quad i \in J \tag{8}$$

$$\eta_{j,t} = f(x_t|S_t = j, \Omega_{t-1}; \theta) = \frac{1}{\sqrt{2\pi}\sigma^{S_t}} \exp\left\{ -\frac{\left(x_t - C^{S_t} - \sum_{k=1}^{P} \varphi_k^{S_t} x_{t-k}\right)^2}{2(\sigma^{S_t})^2} \right\} \tag{9}$$

Obviously, Eq (7) can be derived from Eq (8), Eq (9) and Bayes formula. Thus, the $\theta$ can be obtained by iterative estimation.

Before the application of MS-AR model, the following definitions should be given:

*The regime-switching probability $P_{ij}(i \neq j)$*: the probability of being in regime $j$ at $t$ depends on the regime $i$ at $t-1$, and the probability $P_{ij}(i \neq j)$ of regime $i$ switching to regime $j$ is defined as Eq (10). Then, the regime-switching matrix $P$ can be derived from $P_{ij}(i \neq j)$.

$$P_{ij}(t) = P(S_t = j|S_{t-1} = i) = p_{ij}(t) \tag{10}$$

*The steady-state probability $P_{ij}(i \neq j)$*: different from the regime switching probability $P_{ij}(i \neq j)$, the steady-state probability $P_{ij}(i = j)$ describes the probability that the regime remains unchanged. Then, the steady-state probabilities matrix $Q$ can be derived from $P_{ij}(i = j)$. Since $QP = Q$, i.e., $P^T Q^T = Q^T$, $Q$ can be obtained by finding the eigenvector of $P^T$ with eigenvalue 1.

*The duration $D_i$*: $D_i$ is denoted as the duration that the regime $i$ can last, and satisfies Eq (11):

$$E(D_i) = \sum_{D_i=1}^{\infty} D_i \times P(D_i) = \frac{1}{1 - p_{ij}} \tag{11}$$

**4.2.2. The MS-AR estimation of the proxy variable I.**   Due to the limited space, the MS-AR estimation is detailed in Appendix C in S1 Appendix, and Chongqing is taken as an example to interpret Appendix C in S1 Appendix, as shown in Table 5.

**Table 5.  The MS-AR estimation of the proxy variable I: Chongqing as an example.**

| Regime | Parameter | The proxy variable I | | |
|---|---|---|---|---|
| | | Coefficient | Std. error | Z-statistic |
| Regime 1 | ut(st = 1) | 1.223327*** | 0.049092 | 24.91917 |
| Regime 2 | ut(st = 2) | 0.997505*** | 0.034092 | 29.25894 |
| | | | Regime 1 | Regime 2 |
| Constant Transition Probabilities | | Regime 1 | 0.60986 | 0.39014 |
| | | Regime 2 | 0.153001 | 0.846999 |
| Duration | Regime 1 | 2.563184 | Regime 2 | 6.535899 |

Regime 1 represents the high-risk state, and Regime 2 represents the low-risk state.

*** indicates marginal significance at the 1%-level, ** at the 5% -level, and * at the 10% -level. The sample period is 2010~2018.

It can be seen from Table 5 that: (1) Both the coefficient of high-risk state (1.223327) and the coefficient of low-risk state (0.997505) have passed the 1% significance test, showing that the MS-AR model of Chongqing province is quite reasonable. (2) Although the coefficient of high-risk state is relatively large, the steady-state probability of high-risk state (0.60986) is less than the steady-state probability of low-risk state (0.846999), indicating that Chongqing's local government debt has a relatively strong stability in the low-risk state. (3) The probability of transiting from high-risk state to low-risk state is 0.39014, while the probability from low-risk state to high-risk state is 0.153001, indicating that Chongqing's local government debt changes from a high-risk state to a low-risk state more frequently. In other words, Chongqing's local government debt risk is more likely to ease than worsen in the future. (4) The duration of high-risk state (2.563184) is much less than that of low-risk state (6.535899), reflecting that Chongqing's local government debt lasts longer in the low-risk state.

**4.2.3. The proxy variable II: Individual, contagion, static and dynamic risk.**   Based on the Table 4, Appendices A-C in S1 Appendix, the proxy variable II which comprehensively considers the individual risk, contagion risk, static risk and dynamic risk of local government debt can be obtained by using the coefficient of variation method.

Referring to the practice of Wang and Chen [35], the proxy variable II includes the following seven indicators: the average of the proxy variable I in Table 4; the average of the individual risk index in Appendix A in S1 Appendix; the average of the contagion risk index in Appendix B in S1 Appendix; the steady-state probability of low-risk state, steady-state probability of high-risk state, duration of low-risk state and duration of high-risk state in Appendix C in S1 Appendix.

Before the application of coefficient of variation method, some data processing is needed. Concretely, the above negative indicators should take the "reciprocal" to be consistent with the positive ones. In addition, the seven indicators need to be standardized by the "interval" method, so as to solve the dimension problem.

After data processing, the weights of the seven indicators can be calculated. The results are shown in Table 6. Then, the indicators are multiplied by the corresponding weights to obtain the proxy variable II of each province. The results are shown in Table 7. In Table 7, the larger the proxy variable II of a province, the higher the local government debt risk in the province; and the province with the largest proxy variable II ranks the 1st, indicating that the province is the most risky province of local government debt; et cetera.

In order to show Table 7 more intuitively, China's 30 provinces have been divided into five equal shares according to their risk ranking, as shown in Fig 3. In Fig 3, the five equal shares

**Table 6. Indicators of the proxy variable II.**

| Risk perspectives | Indicators | Descriptions | Attributes | Weights by the coefficient of variation method |
|---|---|---|---|---|
| **Individual Risk + Contagion Risk** | Average of The Proxy Variable I  Z1 | The larger the average of the proxy variable I, the higher the individual risk and contagion risk of local government debt. | Positive | 16.09% |
| | Average of Individual Risk Index  Z2 | The larger the average of individual risk index, the higher the individual risk of local government debt. | Positive | 9.20% |
| | Average of Contagion Risk Index  Z3 | The larger the average of contagion risk index, the higher the contagion risk of local government debt. | Positive | 17.28% |
| **Static Risk + Dynamic Risk** | Steady-state Probability of Low-risk State  Z4 | The larger the steady-state probability of low-risk state, the larger the probability that the local government debt risk will remain in a low-risk state, thus the lower the local government debt risk is. | Negative | 15.65% |
| | Steady-state Probability of High-risk State  Z5 | The larger the steady-state probability of high-risk state, the larger the probability that the local government debt risk will remain in a high-risk state, thus the higher the local government debt risk is. | Positive | 10.54% |
| | Duration of Low-risk State  Z6 | The larger the duration of low-risk state, the longer the local government debt risk will remain in a low-risk state, thus the lower the local government debt risk is. | Negative | 17.03% |
| | Duration of High-risk State  Z7 | The larger the duration of high-risk state, the longer the local government debt risk will remain in a high-risk state, thus the higher the local government debt risk is. | Positive | 14.20% |

correspond to highest-risk, higher-risk, medium-risk, lower-risk and lowest-risk provinces, respectively. It should be emphasized that Fig 3 is the final classification result of this paper, which comprehensively considers the individual risk, contagion risk, static risk and dynamic risk of local government debt. Consequently, Hypothesis 1 is verified.

A comparison between Figs 2 and 3 finds that different risk perspectives lead to different list of provinces that require early warnings: (1) From the perspective of individual risk and contagion risk in Fig 2, regulatory authorities should send different early warning signals to different provinces: high-risk signals to Hunan, Anhui, Tianjin, Shananxi, Hubei, Jiangsu and Guizhou provinces; vulnerability signals to Qinghai, Guangxi, Gansu, Xinjiang, Heilongjiang, Ningxia, Hainan and Yunnan provinces; and high-correlation risk signals to Shandong, Henan, Hebei, Zhejiang, Shanxi, Liaoning and Jilin provinces. (2) From the perspective of individual risk, contagion risk, static risk and dynamic risk in Fig 3, regulatory authorities should send highest-risk signals to Hunan, Jiangsu, Hubei, Henan, Shandong and Shananxi provinces; and higher-risk signals to Anhui, Guangdong, Guizhou, Guangxi, Sichuan and Bei-jing provinces. Therefore, the above different list of risky provinces confirms the Hypothesis 2.

**Table 7. The proxy variable II (individual risk + contagion risk + static risk + dynamic risk).**

| Provinces | Average of the proxy variable I | Average of individual risk index | Average of contagion risk index | Steady-state probability of low-risk state | Steady-state probability of high-risk state | Duration of low-risk state | Duration of high-risk state | The proxy variable II | Risk ranking |
|---|---|---|---|---|---|---|---|---|---|
| Anhui | 1.3404 | 0.8013 | 1.7192 | 0.8833 | 0.8372 | 8.5717 | 6.1429 | 3.1450 | 7 |
| Beijing | 0.8884 | 0.6713 | 1.0410 | 0.8854 | 0.8238 | 8.7295 | 5.6751 | 2.9025 | 12 |
| Chongqing | 1.0730 | 0.8846 | 1.2054 | 0.8470 | 0.6099 | 6.5359 | 2.5632 | 2.1362 | 22 |
| Fujian | 0.7275 | 0.7661 | 0.7004 | 0.6096 | 0.8476 | 2.5616 | 6.5616 | 1.8613 | 25 |
| Gansu | 0.4334 | 0.8249 | 0.1583 | 0.0000 | 0.7420 | 1.0000 | 3.8753 | 0.9718 | 29 |
| Guangdong | 0.9758 | 0.5384 | 1.2832 | 0.8720 | 0.8669 | 7.8148 | 7.5155 | 3.0542 | 8 |
| Guangxi | 0.5205 | 0.8556 | 0.2850 | 0.8744 | 0.8757 | 7.9597 | 8.0435 | 2.9385 | 10 |
| Guizhou | 0.8701 | 0.8680 | 0.8716 | 0.8701 | 0.8747 | 7.6957 | 7.9808 | 3.0427 | 9 |
| Hainan | 0.2805 | 0.6796 | 0.0000 | 0.8365 | 0.8918 | 6.1151 | 9.2416 | 2.6863 | 19 |
| Hebei | 1.3682 | 0.7082 | 1.8321 | 0.8610 | 0.3902 | 7.1926 | 1.6397 | 2.2355 | 21 |
| Heilongjiang | 0.3179 | 0.7704 | 0.0000 | 0.8655 | 0.8823 | 7.4346 | 8.4940 | 2.8227 | 16 |
| Henan | 1.4420 | 0.7385 | 1.9365 | 0.8718 | 0.8759 | 7.7992 | 8.0590 | 3.3359 | 4 |
| Hubei | 1.4705 | 0.7623 | 1.9681 | 0.8849 | 0.8578 | 8.6908 | 7.0315 | 3.3542 | 3 |
| Hunan | 1.5341 | 0.8713 | 1.9998 | 0.8822 | 0.8656 | 8.4918 | 7.4399 | 3.4045 | 1 |
| Inner Mongolia | 0.8519 | 0.8066 | 0.8838 | 0.7917 | 0.8928 | 4.8011 | 9.3302 | 2.7245 | 18 |
| Jiangsu | 1.4783 | 0.7811 | 1.9681 | 0.8851 | 0.8584 | 8.7053 | 7.0641 | 3.3644 | 2 |
| Jiangxi | 0.9040 | 0.7788 | 0.9921 | 0.2756 | 0.5416 | 1.3804 | 2.1816 | 1.0336 | 28 |
| Jilin | 0.4375 | 0.7896 | 0.1900 | 0.8622 | 0.4548 | 7.2594 | 1.8342 | 1.8555 | 26 |
| Liaoning | 0.9615 | 0.8357 | 1.0499 | 0.8713 | 0.0000 | 7.7671 | 1.0000 | 2.0141 | 23 |
| Ningxia | 0.4602 | 0.7595 | 0.2499 | 0.7165 | 0.7158 | 3.5279 | 3.5184 | 1.4751 | 27 |
| Qinghai | 0.3869 | 0.9376 | 0.0000 | 0.8823 | 0.8655 | 8.4960 | 7.4325 | 2.8801 | 13 |
| Shananxi | 1.4122 | 0.8097 | 1.8355 | 0.8349 | 0.8921 | 6.0587 | 9.2653 | 3.1910 | 6 |
| Shandong | 1.4386 | 0.7301 | 1.9365 | 0.8645 | 0.8821 | 7.3792 | 8.4788 | 3.3222 | 5 |
| Shanghai | 0.7377 | 0.4802 | 0.9186 | 0.0000 | 0.0000 | 1.0000 | 1.0000 | 0.6339 | 30 |
| Shanxi | 0.9139 | 0.7092 | 1.0577 | 0.7969 | 0.8953 | 4.9239 | 9.5480 | 2.8085 | 17 |
| Sichuan | 0.6307 | 0.7788 | 0.5267 | 0.8796 | 0.8580 | 8.3056 | 7.0415 | 2.9066 | 11 |
| Tianjin | 1.3905 | 0.9053 | 1.7315 | 0.6859 | 0.7592 | 3.1836 | 4.1528 | 1.9254 | 24 |
| Xinjiang | 0.3353 | 0.8126 | 0.0000 | 0.8741 | 0.8754 | 7.9441 | 8.0284 | 2.8507 | 15 |
| Yunnan | 0.3461 | 0.8386 | 0.0000 | 0.8823 | 0.8654 | 8.4977 | 7.4306 | 2.8644 | 14 |
| Zhejiang | 1.1999 | 0.7273 | 1.5320 | 0.7220 | 0.8812 | 3.5967 | 8.4210 | 2.5389 | 20 |

The sample period is 2010~2018.

**4.2.4. Sensitivity analysis.** A sensitivity analysis is conducted based on changing attribute weights to show the stability of results. To be specific, individual risk and contagion risk index are respectively obtained by the coefficient of variation method. Then, the proxy variable I is obtained by the independent weight method, and the proxy variable II is obtained by the CRITIC method (see S1 Dataset for more details). The results are shown in Appendix D in S1 Appendix.

Comparing the Appendix D in S1 Appendix and Table 7, it is found that the list of provinces receiving early warning signals in Appendix D in S1 Appendix is consistent with that of Table 7 except for Inner Mongolia and Hebei. The comparison demonstrates that changing attribute weights does not have a significant impact on the robustness of the results of this paper.

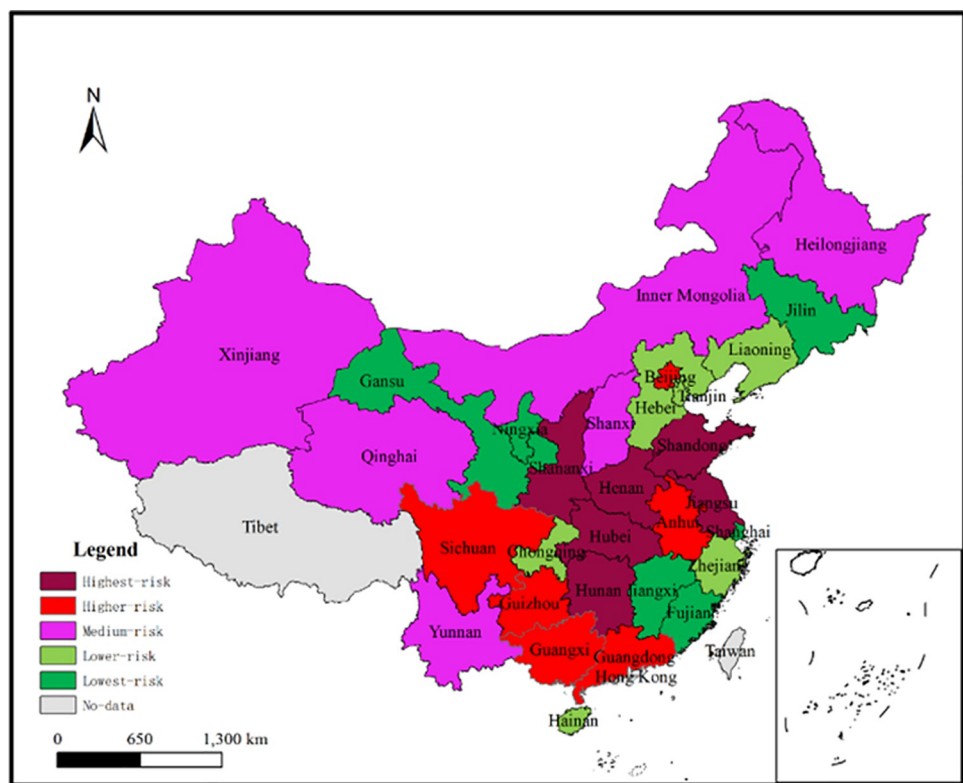

**Fig 3. Five types of local government debt risk: By province.** This is the final classification result: Comprehensive consideration of individual risk, contagion risk, static risk and dynamic risk of local government debt.

## 5. Generalization based on machine learning algorithms

Although the EWS designed in this paper considers the local government debt risk from multiple perspectives, the calculation is complex and cumbersome. It is necessary to improve the generalization ability of our EWS. Therefore, the machine learning algorithms are adopted to debug a classifier to realize a rapid early warning of our EWS.

Machine learning has three algorithms, i.e., supervised learning, unsupervised learning and reinforcement learning. Considering that only provinces with the highest and higher risks will receive early warning signals in this paper, the provinces with the highest and higher risks can thereupon be labeled as 1, and the remaining provinces are labeled as 0. Then, labels 0 and label 1 are used as output variables, and the $X_1$~$X_{11}$ and $Y_1$~$Y_6$ are used as input variables. Obviously, it is a binary classification problem, which is most suitable for using the supervised learning.

Among many algorithms of supervised learning, k-neighbors, linear function, neural network, random forest, gradient boosting machine and support vector machine have been widely used by scholars. Considering China's small sample data, it is necessary to exclude the inappropriate supervised learning algorithms first. According to Andreas and Sarah [61], k-neighbors has slow prediction and cannot deal with a dataset of many feature vectors; linear classifier is limited in low dimensions in face of small sample data; neural network takes a long training time and is easily trapped in the local optimum in the small sample data. However, the random forest, gradient boosting machine and support vector machine have advantages of statistical, computational and representative, without high requirements for sample size [21–23]. Therefore, this paper selects random forest, gradient boosting machine and support vector machine, as detailed in Table 8.

Compared to previous studies, (1) This paper fully considers the characteristics of small sample data of local government debt in China. (2) This paper does not deliberately omit any parameter in the tuning process. (3) This paper uses the machine learning algorithms to generalize the EWS rather than extract experience from experts for the subjective weights. That is, there is no "black box" in this paper, thus maintaining the objectivity of conclusions.

## 5.1. Random forest

Random forest (RF) uses the bootstrap resampling to repeatedly select $n$ samples (usually 2/3) from the original training set $T$ to generate a new training set, each of which is used to train a tree independently. Then, the $n$ decision trees form a forest, in which each tree has the same distribution. Theoretically, the classification error depends on the classification ability of each tree and the correlation between them. In addition, the unselected sample is called out of bag data (OOB), whose error is an unbiased estimate that can be used to validate the performance of the model to prevent overfitting [62].

The generalization error $P^*$ of RF is defined as Eq (12):

$$P^* \leq \frac{\rho(1-s^2)}{s^2} \tag{12}$$

where $\rho$ refers to the average of the correlation of decision trees; $S$ represents the average strength of decision trees. From Eq (12), it is necessary to reduce the correlation between decision trees or increase the strength of decision trees, so as to improve the generalization of RF. To achieve this goal, random disturbance term of feature variables is introduced, resulting in different split nodes of each tree. Finally, the training set and each node variable are randomly generated in the forest.

In the process of growing each classification tree, the splitting of each node is determined by the "purity" of split sample. The "purity" has two criteria, i.e., Gini index and entropy, as shown in Eq (13) and Eq (14), respectively:

$$Gini = 1 - \sum_{i=1}^{k} p_i^2 \tag{13}$$

$$Entropy = -\sum_{i=1}^{k} p_i \log(p_i) \tag{14}$$

where $p_i$ is denoted as the probability of classification $i$. The smaller the Gini index or entropy, the higher the purity of the sample, thus the better the performance of tree splits.

**Table 8. The idea of machine learning classifiers.**

|  | Definitions | Descriptions |
|---|---|---|
| **Output Variables** | 0,1 | The highest-risk and higher-risk provinces are labeled as 1, while the remaining provinces are labeled as 0. |
| **Input Variables** | $X1$:$X11$ and $Y1$:$Y6$ | As shown in Tables 2 and 3. |
| **Test Set** | The data of year 2018 | As shown in Table 7. |
| **Training Set** | The data of year 2016 and 2017 | Taking 2010~2016 and 2010~2017 as the sample period, the proxy variable II and risk ranking of local government debt risk in 2016 and 2017 are calculated respectively. The results are shown in Appendices E and F in S1 Appendix. |
| **Algorithms** | Random Forest, Gradient Boosting Machine, Support Vector Machine | |
| **Software and Version** | Python 3.7.1, Scikit-learn 0.20.1, Numpy 1.15.4, Pandas 0.23.4, Matplotlib 3.0.2, SciPy 1.1.0 | |

## 5.2. Gradient boosting machine

Different from RF, gradient boosting machine (GBM) uses a continuous method to grow decision trees. The algorithm of GBM is as follows [63]:

Suppose that $A = (x_1, x_2, \ldots, x_n)$ serves as the independent variable, $B = \{y_i\}$ refers to the dependent variable and $i = 1, 2, \ldots, n$. For a given dataset, the variables in $A$ need to be mapped to $B$ through a mapping function $f^*(x)$. In addition, the difference between the mapping function and the real function is represented by the loss function $L(y, f(x))$. Obviously, the prediction model should minimize the loss function $L(y, f(x))$ and initialize the mapping function $f^*(x)$ as Eq (15):

$$f^*(x) = \arg \min \sum_{i=1}^{n} L(y_i, \delta) \tag{15}$$

According to the GBM, the direction of $L(y, f(x))$ declines along with the gradient direction, due to the fact that $L(y, f(x))$ changes most significantly in that direction. Thereupon, the negative gradient value of $L(y, f(x))$ is approximately to the residual and can be defined as Eq (16):

$$r_{im} = -\left[\frac{\partial L(y_i, f^*(x_i))}{\partial f^*(x_i)}\right]_{f^*(x) - f^*_{m-1}(x)} \tag{16}$$

The pseudo-residual derived from Eq (16) should be matched with a base classifier $g_m(x)$, which includes various parameters and is trained by the training set $\{(x_i, r_{im})\}_{i=1}^{n}$. Then, the $\delta_m$ can be obtained by the following optimization:

The residual coefficient $y_m$ is derived from Eq (17):

$$y_m = \arg \min \sum_{i=1}^{n} L(y_i, f^*_{m-1}(x_i) + \delta g_m(x_i)) \tag{17}$$

After iteration of $m = 1, 2, \ldots, M$, the optimized prediction function can be obtained, as shown in Eq (18):

$$f^*_m(x) = f^*_{m-1}(x) + \delta_m g_m(x) \tag{18}$$

To sum up, the GBM mainly includes the following parameters: learning rate, tree depth and number of iterations.

## 5.3. Support vector machine

Support vector machine (SVM) attempts to construct an optimal separation hyperplane to classify the positive data and negative data [64]. It starts with mapping the sample data from original space to high-dimensional space through the nonlinear mapping $\varphi(\cdot)$, and optimizes the classification decision function shown in Eq (19). For data points distributed between the classification hyperplane and the support vector, a relaxation variable should be given, with some penalty imposed for the wrong classification. In addition, the optimal hyperplane should satisfy the constraints in Eq (20). Then, the decision function obtained can be expressed as Eq (21), along with the Gaussian radial basis function (RBF) as shown in Eq (22):

$$\min_{\omega, b, \xi} \left\{ \frac{1}{2}\omega^T \omega + C \sum_{i=1} \xi_i \right\} \tag{19}$$

$$y_i(\omega^T \varphi(x_i) + b) \geq 1 - \xi_i \tag{20}$$

$$\text{sgn}(\omega^T \varphi(x) + b) = \text{sgn}[\sum_{i=1} y_i a_i K(x_i, x) + b] \tag{21}$$

$$K(x_i, x) = \exp(- \frac{\|x_i\| - \|x\|^2}{2\gamma^2}) \tag{22}$$

Where $x_i$ refers to the input sample; $y_i$ represents the expected output vector; $\omega$ serves as the weight vector; $C$ is denoted as the penalty parameter and $C \in (0, +\infty)$; $\xi_i$ is defined as the slack variable and $\xi_i \geq 0, (i = 1, 2, \ldots)$; $b$ describes the bias vector; $K(x_i, x)$ is exactly the kernel function, i.e., Gaussian radial basis function; and $a_i$ means the Lagrange multiplier. In brief, $\omega$ and $b$ determine the position of optimal separation hyperplane, and $\gamma$ and $C$ are the parameters that should be optimized.

## 5.4. Performance of RF/GBM/SVM classifiers

This paper uses the 5-fold cross validation and learning curve to debug the RF/GBM/SVM classifiers. The results are shown in Table 9. It can be seen from Table 9 that the roc_auc scores of RF/GBM/SVM classifiers all approach 90%, which are high level in small sample data and exceed their previous roc_auc scores. In addition, all of the three classifiers run very fast. Therefore, regulatory authorities only need to input the original data of $X_1 \sim X_{11}$ and $Y_1 \sim Y_6$ in 2019 and later years into the classifiers, and then the output labeled 1 or 0 will be given by our EWS.

**Table 9. The optimization of RF/GBM/SVM classifiers.**

| Classifiers | Parameters | Descriptions | Optimal values | Previous | Present |
|---|---|---|---|---|---|
| | | | | roc_auc score | roc_auc score |
| **RF Classifier (Running Time: 00:00:013964)** | n_estimators | Number of decision trees | 16 | 0.881944444444444445 | 0.8912037037037037 |
| | max_depth | Maximum depth of decision trees | 1 | | |
| | min_samples_split | Minimum number of samples required to split an internal node | 28 | | |
| | min_samples_leaf | Minimum number of samples required to be at a leaf node | 9 | | |
| | class_weight [a] | Weight of each class | None | | |
| | criterion | Criterion of "purity" | Gini | | |
| **GBM Classifier (Running Time: 00:00:006023)** | n_estimators | Number of continuous trees (weak learner) | 9 | 0.875 | 0.8889 |
| | max_depth | Maximum depth of decision trees | 2 | | |
| | min_samples_split | Minimum number of samples required to split an internal node | 25 | | |
| | subsample | Number of subsamples, i.e, the selected observation of each tree | 0.8 | | |
| | learning_rate | Weight reduction factor of each weak learner | 0.1 | | |
| **SVM Classifier (Running Time: 00:00:010000)** | $\gamma$ | $\gamma$(gamma) represents a parameter of RBF function after it is selected as the kernel. | 2.516527051405394e-05 | 0.694444 | 0.893519 |
| | $C$ | $C$ refers to the penalty strength of the relaxation coefficient, i.e., the tolerance to error. | 0.9269127516778524 | | |

[a] Between "None" and "balanced".

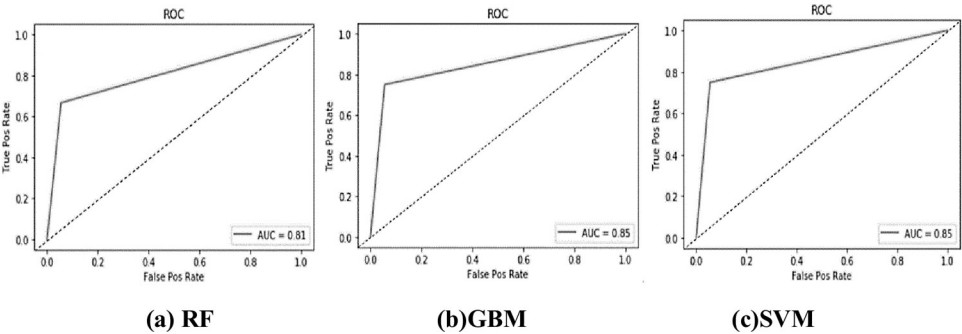

**(a) RF**　　　　　　**(b)GBM**　　　　　　**(c)SVM**

**Fig 4. The ROC curves of RF/GBM/SVM classifiers.** (a) RF (b) GBM (c) SVM.

In order to present the performance of RF/GBM/SVM classifiers more intuitively, this paper provides the ROC curves of the three classifiers, as shown in Fig 4. Obviously, the inflection points of the three ROC curves are all located in the upper left corner, verifying the good performance of the three classifiers again.

Furthermore, the prediction results for the 2018 test set are listed in Table 10. Comparing the true labels with the predicted labels, it is found that both the GBM classifier and SVM

**Table 10. Prediction of RF/GBM/SVM classifiers.**

| Provinces | True label | Predicted label of RF classifier | Predicted label of GBM classifier | Predicted label of SVM classifier |
|---|---|---|---|---|
| Anhui | 1 | 1 | 1 | 1 |
| Beijing | 1 | 1 | 1 | 1 |
| Chongqing | 0 | 0 | 0 | 0 |
| Fujian | 0 | 0 | 0 | 0 |
| Gansu | 0 | 0 | 0 | 0 |
| Guangdong | 1 | 0 | 1 | 0 |
| Guangxi | 1 | 0 | 0 | 0 |
| Guizhou | 1 | 0 | 1 | 1 |
| Hainan | 0 | 0 | 0 | 0 |
| Hebei | 0 | 1 | 1 | 1 |
| Heilongjiang | 0 | 0 | 0 | 0 |
| Henan | 1 | 1 | 1 | 1 |
| Hubei | 1 | 1 | 1 | 1 |
| Hunan | 1 | 1 | 1 | 1 |
| Inner Mongolia | 0 | 0 | 0 | 0 |
| Jiangsu | 1 | 1 | 0 | 1 |
| Jiangxi | 0 | 0 | 0 | 0 |
| Jilin | 0 | 0 | 0 | 0 |
| Liaoning | 0 | 0 | 0 | 0 |
| Ningxia | 0 | 0 | 0 | 0 |
| Qinghai | 0 | 0 | 0 | 0 |
| Shananxi | 1 | 1 | 1 | 1 |
| Shandong | 1 | 1 | 1 | 1 |
| Shanghai | 0 | 0 | 0 | 0 |
| Shanxi | 0 | 0 | 0 | 0 |

(*Continued*)

**Table 10.** (Continued)

| Provinces | True label | Predicted label of RF classifier | Predicted label of GBM classifier | Predicted label of SVM classifier |
|---|---|---|---|---|
| Sichuan | 1 | 0 | 0 | 0 |
| Tianjin | 0 | 0 | 0 | 0 |
| Xinjiang | 0 | 0 | 0 | 0 |
| Yunnan | 0 | 0 | 0 | 0 |
| Zhejiang | 0 | 0 | 0 | 0 |

classifier correctly warn 26 provinces; while the RF classifier only correctly warns 25 provinces. Therefore, the GBM classifier and SVM classifier are the best-performing classifiers in terms of their high prediction accuracy. However, the GBM classifier is inferior to SVM classifier in terms of the roc-auc score, as shown in Table 9. Taken together, the SVM classifier has a better fitting effect on our test set. Thus, the Hypothesis 3 is verified.

## 6. Conclusions and policy recommendations

According to the spatial correlations and spillover effect of China's local government debt risk, this paper constructs a scientific and comprehensive early warning system (EWS) for local government debt risk in China. The data of China's 30 provinces over the period of 2010 to 2018 are taken as a sample data. Through the selection of early warning indicators for individual risk and contagion risk of local government debt, the proxy variable I which combines the two risks are obtained by the CRITIC method and coefficient of variation method for each province. Then, based on the proxy variable I, the proxy variable II which comprehensively considers the individual, contagion, static and dynamic risk of local government debt are estimated by the MS-AR model and coefficient of variation method. Finally, machine learning algorithms are applied to generalize our EWS.

The results show that: (1) From different risk perspectives, the list of provinces that require early warning is different. Specifically, from the individual and contagion risk perspectives, Hunan, Anhui, Tianjin, Shananxi, Hubei, Jiangsu and Guizhou provinces receive high-risk signals; Qinghai, Guangxi, Gansu, Xinjiang, Heilongjiang, Ningxia, Hainan and Yunnan provinces receive vulnerability signals; while Shandong, Henan, Hebei, Zhejiang, Shanxi, Liaoning and Jilin provinces receive high-correlation risk signals. From the individual, contagion, static and dynamic risk perspectives, Hunan, Jiangsu, Hubei, Henan, Shandong and Shananxi provinces receive highest-risk signals; while Anhui, Guangdong, Guizhou, Guangxi, Sichuan and Beijing provinces receive higher-risk signals. (2) The SVM classifier can accurately send early warning signals to highest-risk and higher-risk provinces. Once the original data of $X_1\sim X_{11}$ and $Y_1\sim Y_6$ are input to the SVM classifier, high-accuracy signals can be output, thus realizing a rapid and comprehensive early warning of China's local government debt risk.

Based on the above conclusions, it is recommended that China's regulatory authorities broaden and innovate regulatory ideas, and strengthen the comprehensive supervision of individual risk, contagion risk, static risk and dynamic risk of local government debt.

However, due to data limitations, only ten years of data is available for the analysis, and only the MS-AR model is suitable to investigate the trend of local government debt risk. Further research is expected to focus on the following points: (1) We will keep track of China's data to expand the sample size. (2) Considering that local government debt risk can be studied from multiple perspectives with multiple risk indicators, once the sample size is expanded, we

will build a Markov switching mixed frequency dynamic factor model to characterize the trend of local government debt risk. Then, we can analyze the dynamic transformation path between different risk regimes, thus forming a more in-depth understanding of local government debt risk.

## Supporting information

**S1 Appendix. https://doi.org/10.7910/DVN/1EOZXG.**
(DOCX)

**S1 Dataset. Data and program.** This dataset shows all the data and program in this paper, including those for sensitivity analysis. https://doi.org/10.7910/DVN/OMAQMM.
(RAR)

## Author Contributions

**Conceptualization:** Xing Li, Xiangyu Ge.

**Data curation:** Cong Chen.

**Formal analysis:** Xing Li.

**Investigation:** Xing Li.

**Methodology:** Xing Li, Xiangyu Ge.

**Software:** Xing Li, Cong Chen.

**Supervision:** Xiangyu Ge.

**Visualization:** Cong Chen.

**Writing – original draft:** Xing Li.

**Writing – review & editing:** Xiangyu Ge.

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
