## [Decision Letter · Decision Letter 0]

22 Nov 2021

PONE-D-21-31046Several Explorations on How to Construct an Early Warning System for Local Government Debt Risk in ChinaPLOS ONE

Dear Dr. Li,

Thank you for submitting your manuscript to PLOS ONE. After careful consideration, we feel that it has merit but does not fully meet PLOS ONE’s publication criteria as it currently stands. Therefore, we invite you to submit a revised version of the manuscript that addresses the points raised during the review process.

We look forward to receiving your revised manuscript.

Kind regards,

Mehdi Keshavarz-Ghorabaee

Academic Editor

PLOS ONE

“This research is supported by the National Natural Science Foundation of China (Grant No.71901222 and No. 71974204).”

Reviewers' comments:

Reviewer's Responses to Questions

5. Review Comments to the Author

Reviewer #1: This paper presents a study on a comprehensive early warning system (EWS) for local government debt risk in China. An approach based on the criteria importance though intercrieria correlation (CRITIC) and Markov-switching autoregressive (MS-AR) has been used in this study. I think that the main idea of this paper is interesting. However, I suggest that the authors consider the following comments to improve the paper:

1. I think the justification for using CRITIC and MS-AR should be explained more in the introduction section.

2. I think the paper should be improved by adding some new references to discuss other weighting methods which can be used for the evaluation process. The following approaches should be cited and discussed: Best-Worst Method (BWM), Stepwise Weight Assessment Ratio Analysis (SWARA), SECA (Simultaneous Evaluation of Criteria and Alternatives) and MEREC (MEthod based on the Removal Effects of Criteria).

3. The main features of the previous studies and the current study should be presented in a Table.

4. The structure of the paper should be organized according to the journal requirements.

5. Section headings should be descriptive and concise.

6. You should use some clear diagrams in presentation of the results.

7. A sensitivity analysis should be made based on changing attribute weights to show the stability of results.

Overall, I think the paper needs to be revised before publication.

Reviewer #2: The manuscript studies the local government early warning system, the research methods and tools are relatively good, and the topic has a certain meaning. But the current manuscript still has obvious shortcomings.

1. Lack of a systematic review of the existing literature, this research is not clear from the contribution points in the literature. It is recommended that a separate part be used as a literature review, and on the basis of the literature review, the contribution points of this manuscript are proposed.

2. The paper chooses a provincial-level unit as the research area, which has obvious shortcomings. There have been studies on local government debt and risks in China, usually taking prefecture-level cities and county-level cities as samples. This study takes provincial governments as the research object and is not focused enough. It is recommended that the research object focus on the city and be more targeted.

3. The debts of local governments need to be sorted out in detail. At present, the debts of local governments in China can be divided into statutory debts, with the government acting as the debtor; the other part is implicit debts, which are urban investment bonds issued by local state-owned enterprises. The current debt risk is mainly concentrated in implicit debt. Whether the local government debt in this study includes two aspects or only includes statutory debt, it needs to be further clarified.

4. Generally speaking, the paper has a relatively large workload and the tools and methods are relatively reasonable, but the literature review and contribution are not clear yet.

Reviewer #3: EVALUATION ÄPER PONE-D-21-31046

1. The paper is too long 37 pages.

2. Question research is clear and intersting

3. Globally, the literature review is rich and up-to-date

4. Tiltles should be improved: too long and sometimes very general such as: algorithm p33

5.

Construction ideas: this section have to be improved and restructured (page 12)

• Conceptuel framework and theory

• Litterature review

• Hypothesis

Indicator data and methods : Methodology or research design:

It is preferable to change titles: Definition 1,2,3,4,5…. : tiltles must be improved

6. It is not a good idea to insert tables or graphs in the conclusion

Check the references inserted in the text : there are some references which do not appear in the end of the paper

7. Hypothesis are not very clear and not well formulated

8. There is not any appendices but in the text author inserted in page 20; 21;26………

---

## [Author Response · Author response to Decision Letter 0]

3 Jan 2022

Response to the comments from reviewer #1

Introduction 

Comment: I think the justification for using CRITIC and MS-AR should be explained more in the introduction section. 

Response: We thank the reviewer for the reminder. As suggested, we add the following content in the third paragraph of the revised “Introduction” to explain the use of CRITIC (the criteria importance though intercrieria correlation) and MS-AR:

The marginal contributions of this paper are as follows: (1)…(3) Rather than using the subjective weighting methods, the objective weighting methods (the criteria importance though intercrieria correlation method and coefficient of variation method) will be used to determine the weight of early warning indicators, thus ensuring the fairness of our indicator weighting. (4) Considering that the data of local government debt risk of China's each province are typically non-linear time series and the Markov-switching autoregressive (MS-AR) model can effectively capture different regimes of time series data [10-12], the MS-AR model will be used to study the static risk and dynamic risk of local government debt for each province.

Methodology

Comment: I think the paper should be improved by adding some new references to discuss other weighting methods which can be used for the evaluation process. The following approaches should be cited and discussed: Best-Worst Method (BWM), Stepwise Weight Assessment Ratio Analysis (SWARA), SECA (Simultaneous Evaluation of Criteria and Alternatives) and MEREC (MEthod based on the Removal Effects of Criteria). 

Response: This is a great point. The suggested change has been made in the first paragraph of revised “4.1.3. Individual risk index and contagion risk index”:

There exists various weighting methods, i.e., the subjective weighting represented by AHP, the objective weighting represented by CRITIC method and coefficient of variation method, the subjective-objective integrated weighting, and the newly-emerging weighting represented by BWM (Best-Worst Method) [56], SWARA (Simultaneous Evaluation of Criteria and Alternatives) [57], SECA (Simultaneous Evaluation of Criteria and Alternatives) [58] and MEREC (Method Based on the Removal Effects of Criteria) [59]. Considering that subjective weighting incorporates subjective judgments of decision maker while some objective weighting needs fussy work with much calculation, this paper adopts the CRITIC method and coefficient of variation method to give objective weights to indicators.

And we add the following references:

56. Rezaei J. Best-worst Multi-criteria Decision-making Method. Omega. 2015; 53:49-57.

57. Kersuliene V, Zavadskas EK, Turskis Z. Selection of Rational Dispute Resolution Method by Applying New Step-wise Weight Assessment Ration Analysis (SWARA). Journal of Business Economics and Management. 2010; 11(2): 243-258.

58. Keshavarz-Ghorabaee M, Amiri M, Zavadskas EK, Turskis Z, Antucheviciene J. Simultaneous Evaluation of Criteria and Alternatives (SECA) for Multi-Criteria Decision-Making. Informatica. 2018; 29(2):265-280.

59. Keshavarz-Ghorabaee M, Amiri M, Zavadskas EK, Turskis Z, Antucheviciene J. Determination of Objective Weights Using a New Method Based on the Removal Effects of Criteria (MEREC). Symmetry. 2021; 13: 525.

Literature review

Comment: The main features of the previous studies and the current study should be presented in a Table.

Response: We are grateful for the comments made by reviewer. As suggested, “Table 1. The previous EWSs and our EWS.” has been added to the revised manuscript:

Table 1. The previous EWSs and our EWS. Details can be seen in the revised manuscript.

Comment: The structure of the paper should be organized according to the journal requirements.

Response: We thank the reviewer for the reminder. As suggested, 

1. We split the original "1. Introduction" into two parts: "1.Introduction" and "2.Literature review". 

2. We reorganize the original "2. Construction Ideas" and split it into "3.1. Conceptual framework", "3.2. Hypothesis" and "3.3. Research design" (as requested by the reviewer #3). 

3. We change the original "3. Indicators, Data and Methods" to "4. Methodology", and split it into "4.1. The proxy variable Ⅰ" and "4.2. The proxy variable Ⅱ", and supplement "4.2.4. Sensitivity" analysis" to verify the robustness of the results of this paper. 

4. We simply the structure of the original "4. Generalization based on Machine Learning Algorithms". To be specific, we only set the secondary headings, which are "5.1. Random forest", "5.2. Gradient boosting machine", "5.3. Support vector machine" and "5.4. Performance of RF/GBM/SVM classifiers". 

5. We simply the original "5. Conclusions and Policy Recommendations" and turn the conclusion in "Table 10" into text description, details are in the revised manuscript.

(Notes:The value A= The proxy variable Ⅰ, The value B= The proxy variable Ⅱ)

The reorganized structure is as follows:

1.Introduction

2. Literature review

3. Construction ideas

3.1. Conceptual framework

3.2. Hypothesis

3.3. Research design

4. Methodology

4.1. The proxy variable Ⅰ

4.1.1. Early warning indicators of individual risk

4.1.2. Early warning indicators of contagion risk

4.1.3. Individual risk index and contagion risk index

4.1.4. The proxy variable Ⅰ: individual and contagion risk

4.2. The proxy variable Ⅱ

4.2.1. The MS-AR model

4.2.2. The MS-AR estimation of the proxy variable Ⅰ

4.2.3. The proxy variable Ⅱ: individual, contagion, static and dynamic risk

4.2.4. Sensitivity analysis

5. Generalization based on machine learning algorithms

5.1. Random forest

5.2. Gradient boosting machine

5.3. Support vector machine

5.4. Performance of RF/GBM/SVM classifiers

6. Conclusions and policy recommendations

Comment: Section headings should be descriptive and concise.

Response: We thank the reviewer for the reminder. As suggested, we try our best to simplify the section headings as much as possible. For example:

1. We simply the original "3. Indicators, Data and Methods" to "4. Methodology".

2. We simply the original "3.1. The value A: individual risk plus contagion risk" to "4.1. The proxy variable Ⅰ".

3. We delete the "by the CRITIC method" from the original "3.1.3. Individual risk index and contagion risk index by the CRITIC method".

4. We change the original "3.1.4. The value A by the coefficient of variation method" to "4.1.4. The proxy variable Ⅰ: individual and contagion risk".

5. We simply the original "3.2. The value B: individual risk plus contagion risk plus static risk plus dynamic risk" to "4.2. The proxy variable Ⅱ".

6. We change the original "3.2.3. The value B" to "4.2.3. The proxy variable Ⅱ: individual, contagion, static and dynamic risk".

7. We only set the secondary headings in the original "4. Generalization based on Machine Learning Algorithms", i.e., "5.1. Random forest", "5.2. Gradient boosting machine", "5.3. Support vector machine" and "5.4. Performance of RF/GBM/SVM classifiers". 

(Notes: The value A= The proxy variable Ⅰ, The value B= The proxy variable Ⅱ)

Comment: You should use some clear diagrams in presentation of the results.

Response: We are grateful for the comments made by reviewer. As suggested, we changed the original Fig. 2 into a map to make it clearer.

The same change has been made to the original Fig. 3.

Details are in the revised manuscript.

Methodology

Comment: A sensitivity analysis should be made based on changing attribute weights to show the stability of results.

Response: We are grateful for the comments made by reviewer. As suggested, a sensitivity analysis has been added in the revised manuscript "4.2.4. Sensitivity" analysis". We write:

A sensitivity analysis is conducted based on changing attribute weights to show the stability of results. To be specific, individual risk and contagion risk index are respectively obtained by the coefficient of variation method. Then, the proxy variable Ⅰ is obtained by the independent weight method, and the proxy variable Ⅱ is obtained by the CRITIC method. The results are shown in Appendix D. 

Comparing the Appendix D and Table 7, it is found that the list of provinces receiving early warning signals in Appendix D is consistent with that of Table 7 except for Inner Mongolia and Hebei. The comparison demonstrates that changing attribute weights does not have a significant impact on the robustness of the results of this paper.

The "Appendix D. Sensitivity analysis" can be seen in the revised Appendix.

The calculation can be downloaded from:

https://doi.org/10.7910/DVN/OMAQMM

Response to the comments from reviewer #2

Literature review

Comment: Lack of a systematic review of the existing literature, this research is not clear from the contribution points in the literature. It is recommended that a separate part be used as a literature review, and on the basis of the literature review, the contribution points of this manuscript are proposed.

Response: We are grateful for the comments made by reviewer. As suggested, we split the original "1. Introduction" into two parts: "1. Introduction" and "2.Literature review". 

In the "2. Literature review", we have combed the literature on the construction of EWS by Western scholars and Chinese scholars. In addition, we add the “Table 1. The previous EWSs and our EWS.” as suggested by the reviewer #1. Then, we point out the research gap of the existing literature, and explain the research content of this paper. The revisions can be seen in the "2. Literature review":

As western countries granted governments the right to raise debt earlier than China, western scholars made pioneering attempts in constructing the EWS. The study on the EWS for public debt can be tracked back to 1980s [13], and has sprung up in the US, Italy and Colombia. Recently, …

Among the existing EWSs constructed by western scholars, the multinomial logistic regression belonging to the econometric models has been the most widely used. Pioneering studies by Ciarlone and Trebeschi [14]…

With the development of software and econometric models, a new trend in the latest work is to introduce the machine learning algorithms into the EWS for municipal debt risk. Representatives are Antulov-Fantulin et al. [21], Alaminos et al. [22] and Zahariev et al. [23]. To be specific,…

Faced with less than 15 years of local government debt data, Chinese scholars still made valuable explorations on the construction of EWS. Representatives are the BP neural network by Shi et al. [24] and Hong and Liu [25],…

Table 1. The previous EWSs and our EWS...

Although previous studies have great enlightenments in constructing EWS for public debt risk, there are still some research gaps at the local government debt level, especially for China's local government debt risk. The research gaps are as follows: (1) …

In view of the above research gaps, this paper intends to construct an EWS from multiple perspectives of local government debt risk, and debugs a machine learning classifier to generalize the EWS. We believe that our EWS…

In addition, we propose the marginal contributions of this paper in the third paragraph of the revised "1. Introduction":

The marginal contributions of this paper are as follows: (1) Besides the explicit local government debt, the implicit local government debt will be included in this paper, making our EWS cover more types of local government debt. (2) Based on the spatial spillover effect of China's local government debt risk [7-9], the contagion risk of local government debt will not be ignored in our EWS. (3) Rather than using the subjective weighting methods, the objective weighting methods …

Methodology

Comment: The paper chooses a provincial-level unit as the research area, which has obvious shortcomings. There have been studies on local government debt and risks in China, usually taking prefecture-level cities and county-level cities as samples. This study takes provincial governments as the research object and is not focused enough. It is recommended that the research object focus on the city and be more targeted.

Response: The comments made by reviewer are very constructive, and we fully agree with the reviewer. 

However, due to the limited data access, only few research teams in China can compile and obtain county and municipal data of local government debt. The representative is a research team led by Professor Mao Jie from the University of International Business and Economics, e.g., 

50. Mao J, Huang CY. Local Debts, Regional Disparity and Economic Growth: An Empirical Study Based on China's Prefecture -Level Data. Journal of Financial Research. 2018; 5:1-19. 

Due to limited researchers and data access, we did not sort out county and city-level data of China's local government debt. Instead, we sorted out the provincial data of China's local government debt based on the Wind database. We write in the revised manuscript: 

4. Methodology

Due to the availability of data, this paper chooses a provincial-level unit as the research area [45-48]. Additionally,…

We believe that provincial data are also still representative, meaningful, and widely used by scholars, e.g.

45. Chen Z, He ZG, Liu C. The Financing of Local Government in China: Stimulus Loan Wanes and Shadow Banking Waxes. NBER Working Paper Series. 2018; 23598.

46. Zhao RB, Tian YX, Lei A, Boadu F, Ren Z. The Effect of Local Government Debt on Regional Economic Growth in China: A Nonlinear Relationship Approach. Sustainability. 2019;11(11), 3065.

47. Ouyang TH, Lu XY. Clustering Analysis of Risk Divergence of China Government’s Debts. Scientific Programming. 2021; 7033597.

48. Zheng CJ, Huang SW, Qian NY. Analysis of the Co-movement Between Local Government Debt Risk and Bank Risk in China. The Singapore Economic Review. 2021; 66(03): 807-835.

We are grateful for the comments made by reviewer. In the future, we will refer to the comments made by reviewer and expand our research after we obtain the county and city-level data.

Comment: The debts of local governments need to be sorted out in detail. At present, the debts of local governments in China can be divided into statutory debts, with the government acting as the debtor; the other part is implicit debts, which are urban investment bonds issued by local state-owned enterprises. The current debt risk is mainly concentrated in implicit debt. Whether the local government debt in this study includes two aspects or only includes statutory debt, it needs to be further clarified.

Response: We thank the reviewer for the reminder. As suggested, we add a beginning paragraph in the revised “4. Methodology”:

Due to the availability of data, this paper chooses a provincial-level unit as the research area [45-48]. Additionally, in order to cover as many types of local government debt as possible, this paper classifies the local government debt into explicit debt and implicit debt according to Polackova's definitions [49]. Then, referring to the statistical caliber of Mao and Huang [50] and Wang [51], the explicit debt is composed of local government bonds and debt re-loans, and the implicit debt is composed of urban investment bond.

Comment: Generally speaking, the paper has a relatively large workload and the tools and methods are relatively reasonable, but the literature review and contribution are not clear yet.

Response: We are grateful for the comments made by reviewer. As suggested, we split the original "1. Introduction" into two parts: "1. Introduction" and "2. Literature review", and made many revisions in the "2. Literature review", as shown in the revised manuscript.

In addition, we propose the marginal contributions of this paper in the third paragraph of the revised "1. Introduction":

The marginal contributions of this paper are as follows: (1) Besides the explicit local government debt, the implicit local government debt will be included in this paper, making our EWS cover more types of local government debt. (2) Based on the spatial spillover effect of China's local government debt risk [7-9], the contagion risk of local government debt will not be ignored in our EWS. (3) Rather than using the subjective weighting methods, the objective weighting methods …

Response to the comments from reviewer #3

Introduction

Comment: The paper is too long 37 pages.

Response: We are grateful for the comments made by reviewer. We have tried our best to delete some lengthy paragraphs from the original manuscript. The word count (including the table) is compressed to less than 13,000 words.

In order to clarify the research methods and process, some Tables and Figures cannot be deleted. Therefore, we recommend storing these Tables and Figures as network resources in PLOS ONE. Please see the "Response to Reviewers" for the template.

We hope our recommendation can get your understanding.

Comment: Question research is clear and interesting.

Response: Thank the comments made by reviewer. We will continue to work hard to improve this paper.

Comment: Globally, the literature review is rich and up-to-date.

Response: Thank the comments made by reviewer. We still make the following revisions: 

1. We have split the original "1. Introduction" into two parts: "1. Introduction" and "2. Literature review".

2. In the "2. Literature review", we have combed the literature on the construction of EWS by Western scholars and Chinese scholars, and added the “Table 1. The previous EWSs and our EWS” as suggested by the reviewer #1. Then, we have pointed out the research gap of the existing literature, and explained the research content of this paper. The revisions can be seen in the "2. Literature review".

Comment: Titles should be improved: too long and sometimes very general such as: algorithm p33.

Response: We thank the reviewer for the reminder. As requested, we have simplified the titles as much as possible. The revisions are as follows:

1. We simply the original "3. Indicators, Data and Methods" to "4. Methodology".

2. We simply the original "3.1. The value A: individual risk plus contagion risk" to "4.1. The proxy variable Ⅰ".

3. We delete the "by the CRITIC method" from the original "3.1.3. Individual risk index and contagion risk index by the CRITIC method".

4. We change the original "3.1.4. The value A by the coefficient of variation method" to "4.1.4. The proxy variable Ⅰ: individual and contagion risk".

5. We simply the original "3.2. The value B: individual risk plus contagion risk plus static risk plus dynamic risk" to "4.2. The proxy variable Ⅱ".

6. We change the original "3.2.3. The value B" to "4.2.3. The proxy variable Ⅱ: individual, contagion, static and dynamic risk".

7. We only set the secondary headings in the original "4. Generalization based on Machine Learning Algorithms", which are "5.1. Random forest", "5.2. Gradient boosting machine", "5.3. Support vector machine" and "5.4. Performance of RF/GBM/SVM classifiers". 

(Notes: The value A= The proxy variable Ⅰ, The value B= The proxy variable Ⅱ)

Construction ideas

Comment: Construction ideas: this section have to be improved and restructured (page 12)

• Conceptual framework and theory • Literature review • Hypothesis

➢ Indicator data and methods : Methodology or research design: 

It is preferable to change titles: Definition 1,2,3,4,5…. : titles must be improved

Response: We are grateful for the comments made by reviewer. As suggested, we have restructured this section. The revised structure is as follows:

2. Literature review

3. Construction ideas

3.1. Conceptual framework

3.2. Hypothesis

3.3. Research design

4. Methodology

…

Details can be seen in the revised manuscript.

Moreover, we have deleted the original “Definition 1,2,3,4,5…”. Instead, we showed these concepts in italics (details can be seen in the revised manuscript):

3.1. Conceptual framework

There are four types of local government debt risks in this paper:

The individual risk of local government debt: due to the fact that local government as a sub-national administrative agency cannot easily go bankruptcy, the…

The contagion risk of local government debt: it can be defined as "too tightly correlated to fail" of financial risk. Referring to the empirical…

The static risk of local government debt: it refers to how risky the local government debt is when it maintains its original risk state…

The dynamic risk of local government debt: it describes the regime-switching risk of local government debt when it switches…

 ……

4.2.1. The MS-AR model

The regime-switching probability: the probability of being in regime at depends on the regime at …

The steady-state probability : different from the regime switching probability …

The duration : is denoted as the duration that the regime can last, and satisfies Eq ([Disp-formula pone.0263391.e034]) …

Conclusions and policy recommendations

Comment: It is not a good idea to insert tables or graphs in the conclusion

Check the references inserted in the text: there are some references which do not appear in the end of the paper

Response: We thank the reviewer for the reminder. As suggested, we have deleted the original “Table 10”, and described it as follows:

The results show that: (1) From different risk perspectives, the list of provinces that require early warning is different. Specifically, from the individual and contagion risk perspectives, Hunan, Anhui, Tianjin, Shananxi, Hubei, Jiangsu and Guizhou receive high-risk signals; Qinghai, Guangxi, Gansu, Xinjiang, Heilongjiang, Ningxia, Hainan and Yunnan receive vulnerability signals; while Shandong, Henan, Hebei, Zhejiang, Shanxi, Liaoning and Jilin receive high-correlation risk signals. From the individual, contagion, static and dynamic risk perspectives, Hunan, Jiangsu, Hubei, Henan, Shandong and Shananxi receive highest-risk signals; while Anhui, Guangdong, Guizhou, Guangxi, Sichuan and Beijing receive higher-risk signals. (2) The SVM classifier can accurately send early warning signals…

In addition, we have checked the citations of the literature one by one, and deleted the un-cited literature in the revised “References”. Details can be seen in the revised manuscript.

Construction ideas

Comment: Hypothesis are not very clear and not well formulated.

Response: We are grateful for the comments made by reviewer. As requested, we have added a section “3.2. Hypothesis” in the revised manuscript. We write:

3.2. Hypothesis

Risks are not single and static, but contagious and dynamic. The contagion effect and dynamics can be described by empirical models. Regarding the contagion risk, Bianchi et al. [38] take the network structure perspective…

Hypothesis 1: A proxy variable that comprehensively considers the individual risk, contagion risk, static risk and dynamic risk of local government debt of each province can be obtained by...

In risk research, there is a heated discussion on foundational issues about concepts and perspectives. The development of well-founded risk perspectives is a crucial issue to intensify the practice of risk analysis. Fundamentally, …

Hypothesis 2: Different risk perspectives may lead to different list of provinces that require early warnings.

Moreover, due to the numerous advances in software, recent works increasingly use the machine learning algorithms to finalize…

Hypothesis 3: An optimal machine learning classifier can be debugged to generalize our EWS.

Comment: There is not any appendices but in the text author inserted in page 20; 21;26…….

Response: We feel guilty of not showing the appendix. As suggested, we have saved all the appendices at

https://doi.org/10.7910/DVN/1EOZXG

and pointed out the link in the revised manuscript.

As displayed above, we have tried our best to improve the manuscript and made some changes in the manuscript. 

Once again, we appreciate for Editor and Reviewers’ valuable work and hope that the correction will meet with approval.

Best wishes,

Sincerely yours,

Xing Li, Xiangyu Ge and Cong Chen

---

## [Editor Report · Decision Letter 1]

19 Jan 2022

Several Explorations on How to Construct an Early Warning System for Local Government Debt Risk in China

PONE-D-21-31046R1

Dear Dr. Li,

We’re pleased to inform you that your manuscript has been judged scientifically suitable for publication and will be formally accepted for publication once it meets all outstanding technical requirements.

Kind regards,

Mehdi Keshavarz-Ghorabaee

Academic Editor

PLOS ONE

---

## [Editor Report · Acceptance letter]

28 Jan 2022

PONE-D-21-31046R1 

Several Explorations on How to Construct an Early Warning System for Local Government Debt Risk in China 

Dear Dr. Li:

I'm pleased to inform you that your manuscript has been deemed suitable for publication in PLOS ONE. Congratulations! Your manuscript is now with our production department. 

Kind regards, 

on behalf of

Dr. Mehdi Keshavarz-Ghorabaee 

Academic Editor

PLOS ONE